# HERTA: A High-Efficiency and Rigorous Training Algorithm for Unfolded Graph Neural Networks

## Abstract

As a variant of Graph Neural Networks (GNNs), Unfolded GNNs offer enhanced interpretability and flexibility over traditional designs. Nevertheless, they still suffer from scalability challenges when it comes to the training cost. Although many methods have been proposed to address the scalability issues, they mostly focus on per-iteration efficiency, without worst-case convergence guarantees. Moreover, those methods typically add components to or modify the original model, thus possibly breaking the interpretability of Unfolded GNNs. In this paper, we propose HERTA: a High-Efficiency and Rigorous Training Algorithm for Unfolded GNNs that accelerates the whole training process, achieving a nearly-linear time worst-case training guarantee. Crucially, HERTA converges to the optimum of the original model, thus preserving the interpretability of Unfolded GNNs. Additionally, as a byproduct of HERTA, we propose a new spectral sparsification method applicable to normalized and regularized graph Laplacians that ensures tighter bounds for our algorithm than existing spectral sparsifiers do. Experiments on real-world datasets verify the superiority of HERTA as well as its adaptability to various loss functions and optimizers.

## 1 Introduction

Graph Neural Networks (GNNs) have become a powerful modern tool for handling graph data because of their strong representational capabilities and ability to explore the relationships between data points Kipf & Welling (2016); Battaglia et al. (2018); Xu et al. (2018). Like many deep learning models, GNNs are generally designed by heuristics and experience, which makes analyzing and understanding them a difficult task. *Unfolded GNNs* Yang et al. (2021a) are a type of GNNs that are rigorously derived from an optimization problem, which makes their inductive bias and training dynamics easier to interpret.

Despite their enhanced interpretability, training unfolded models can be extremely expensive, especially when the graph is large and (relatively) dense. This issue can come from two aspects:

1) *Slow iterations*: in every iteration the whole graph must be fed into the model, and the interactive nature of graphs prevents trivially utilizing techniques like mini-batch training;

2) *Slow convergence*: the training convergence rate is related to the connectivity of the graph and the well-conditionedness of the node features, and the model converges slowly when the data is ill-conditioned.

Many methods have been proposed to address the high cost of training unfolded models Chen et al. (2018a); Chiang et al. (2019); Zeng et al. (2021); Ding et al. (2021); Fey et al. (2021); Jiang et al. (2023), primarily by using graph sampling to enable mini-batch training schemes that reduce the per-iteration cost (Issue 1). However, these approaches typically require distorting the underlying optimization objective explicitly or implicitly, thus diminishing the rigorous and interpretable nature of Unfolded GNNs. Moreover, existing works have mostly focused on per-iteration efficiency, while the convergence rate of the optimization in training Unfolded GNNs (Issue 2) remains un-addressed, leading to methods that are not robust to ill-conditioned data.

We propose HERTA: a High-Efficiency and Rigorous Training Algorithm for Unfolded GNNs, which is an algorithmic framework designed to address both Issues 1 and 2, while at the same

time preserving the rigorous and interpretable nature of Unfolded GNNs. Unlike many existing methods that require changing the GNN model or defining a new objective, HERTA converges to the optimum of the original training problem, and thus preserves the interpretability of the unfolded models. Moreover, HERTA uses a specialized preconditioner to ensure fast linear convergence to the optimum, requiring only a logarithmic number of passes over the data. We show empirically that HERTA can be used to accelerate training for a variety of GNN objectives, as an extension of popular optimization techniques.

A key novelty of our algorithm lies in the construction of a preconditioner for the GNN objective, which accelerates the convergence rate. To construct this preconditioner, we design a spectral sparsification method for approximating the squared inverse of a normalized and regularized Laplacian matrix, by relying on an extension of the notion of effective resistance to overcome the limitations of existing graph sparsification tools. Below, we present an informal statement of our main result.

**Theorem 1.1** (Informal version of Theorem 5.1). *HERTA solves the $\lambda$-regularized Unfolded GNN objective equation 4.2 with $n$ nodes, $m$ edges and $d$-dimensional node features to within accuracy $\epsilon$ in time $\tilde{O}\left((m + nd)\left(\log \frac{1}{\epsilon}\right)^2 + d^3\right)$ as long as the number of large eigenvalues of the graph Laplacian is $O(n/\lambda^2)$.*

In practice, the node feature dimensionality $d$ is generally much smaller than the graph size $m$, in which case, the running time of HERTA is $\tilde{O}\left((m + nd)\left(\log \frac{1}{\epsilon}\right)^2\right)$. Notice that to describe a typical graph dataset with $m$ edges, $n$ nodes and $d$-dimensional node features, we need at least $O(m + nd)$ float or integer numbers. This shows the essential optimality of HERTA: its running time is of the same magnitude as reading the input data (up to logarithm factors).

The condition that the graph Laplacian has $O(n/\lambda^2)$ large eigenvalues is not strictly necessary; it is used here to simplify the time complexity (see Theorem 5.1 for the full statement). This condition is also not particularly restrictive, since in most practical settings the GNN parameter $\lambda$ is chosen as an absolute constant. At the same time, it is natural that the complexity of training a GNN depends on the number of large eigenvalues in the graph Laplacian, since this quantity can be interpreted as the effective dimension in the Laplacian energy function equation 4.1 used to define the Unfolded GNN (see more discussion in Section 5).

**Outline.** The paper is organized as follows. In Section 2 we introduce related work. In Section 3 we introduce the mathematical notation and concepts used in this paper. We define our problem setting in Section 4. In Section 5 we present and analyze our algorithm HERTA, and introduce the techniques that are used in proving the main result. We conduct experiments on real-world datasets and show the results in Section 6. Finally, we conclude with some potential future directions in Section 7.

## 2  RELATED WORK

**Unfolded Graph Neural Networks.** Unlike conventional GNN models, which are designed mainly by heuristics, the forward layers of Unfolded GNNs are derived by explicitly optimizing a graph-regularized target function Zhang et al. (2020); Yang et al. (2021a); Ma et al. (2021); Liu et al. (2021); Yang et al. (2021b); Zhu et al. (2021); Ahn et al. (2022). This optimization nature of Unfolded GNNs allows developing models in a more interpretable and controllable way. The presence of an explicit optimization problem allows designers to inject desired properties into the model and better understand it. This approach has been used to overcome known GNN issues such as over-smoothing Oono & Suzuki (2019); Liu et al. (2020); Yang et al. (2020) and sensitivity towards spurious edges Zügner et al. (2018); Zhu et al. (2020).

**Efficient Graph Neural Network Training.** In order to address the scalability issue of GNNs, many techniques have been proposed. There are two major directions in the exploration of the solutions of this issue. The first direction is to adopt sampling methods. These methods include node-wise sampling Hamilton et al. (2017), layer-wise sampling Chen et al. (2018b) and subgraph-wise sampling Chiang et al. (2019); Jiang et al. (2023). The second direction is to utilize embeddings from past iterations to enhance current iterations and to obtain more accurate representation with mini-batches and fewer forward layers Fey et al. (2021); Ding et al. (2021); Xue et al. (2023).

As we indicated in Section 1, all of the above methods aim at addressing Issue 1 (*Slow Iterations*), and none of them achieves a worst-case convergence guarantee without affecting the underlying GNN model. In contrast, HERTA addresses both Issue 1 and also Issue 2 (*Slow Convergence*), and possesses a theoretical guarantee on the training convergence rate while preserving the original target model.

**Matrix Sketching and Subsampling.**  Our techniques and analysis are closely related to matrix sketching and subsampling, which are primarily studied in the area of Randomized Numerical Linear Algebra (RandNLA, Woodruff et al. (2014); Drineas & Mahoney (2016); Martinsson & Tropp (2020); Murray et al. (2023)). Given a large matrix, by sketching or sampling we compress it to a much smaller matrix with certain properties preserved, thus accelerating the solution by conducting expensive computations on the smaller matrix. This type of methods lead to improved randomized algorithms for tasks including low rank approximation Halko et al. (2011); Cohen et al. (2017a); Clarkson & Woodruff (2017), linear system solving Peng & Vempala (2021); Dereziński & Yang (2024), least squares regression Rokhlin & Tygert (2008); Meng et al. (2014) and so on Cohen et al. (2021). Our usage of these methods includes constructing spectral sparsifiers of graphs, obtained by edge subsampling Spielman & Srivastava (2008); Spielman & Teng (2011); Koutis et al. (2016), which are central in designing fast solvers for Laplacian linear systems Vishnoi et al. (2013); Kelner et al. (2013); Peng & Spielman (2014). We note that constructing spectral sparsifiers with regularization has also been explored in Calandriello et al. (2018), however their setting is different from ours: we consider the *normalized* Laplacian of a graph, which is not a Laplacian of any graph (even allowing weighted edges).

## 3    PRELIMINARIES

For a vector $\boldsymbol{x}$, we denote its $\ell_2$ norm as $\|\boldsymbol{x}\|$. We use $\mathrm{diag}(\boldsymbol{x})$ to denote diagonalization of $\boldsymbol{x}$ into a matrix. For a matrix $\boldsymbol{M}$, we use $\|\boldsymbol{M}\|_{\mathcal{F}}$ and $\|\boldsymbol{M}\|$ to denote its Frobinius norm and operator norm. Let $\sigma_{\max}(\boldsymbol{M})$ and $\sigma_{\min}(\boldsymbol{M})$ be the largest and smallest singular value of $\boldsymbol{M}$ respectively, we denote its condition number as $\kappa(\boldsymbol{M}) = \frac{\sigma_{\max}(\boldsymbol{M})}{\sigma_{\min}(\boldsymbol{M})}$. We use $\mathrm{nnz}(\boldsymbol{M})$ to denote the number of non-zero entries of $\boldsymbol{M}$. For two positive semidefinite (PSD) matrices $\boldsymbol{\Sigma}$ and $\widetilde{\boldsymbol{\Sigma}}$, if there exists $\epsilon \in (0,1)$ such that $(1-\epsilon)\widetilde{\boldsymbol{\Sigma}} \preceq \boldsymbol{\Sigma} \preceq (1+\epsilon)\widetilde{\boldsymbol{\Sigma}}$, then we say $\boldsymbol{\Sigma} \approx_\epsilon \widetilde{\boldsymbol{\Sigma}}$, where $\preceq$ refers to Loewner order. We use $\boldsymbol{\delta}_u$ to denote a vector with $u$-th entry equal to 1 and all other entries equal to 0. We also use $\delta_{u,v}$ to represent the $v$-th entry of $\boldsymbol{\delta}_u$, i.e. $\delta_{u,v} = \begin{cases} 1 & u = v \\ 0 & u \neq v \end{cases}$.

For a function $\ell : \mathbb{R}^d \to \mathbb{R}$ that is bounded from below, a point $\boldsymbol{w}_0$ is called a solution of $\ell$ with $\epsilon$ error rate if it satisfies $\ell(\boldsymbol{w}_0) \leq (1+\epsilon)\ell^*$ where $\ell^* = \inf_{\boldsymbol{w}} \ell(\boldsymbol{w})$.

We adopt big-O notations in the time complexity analysis. Moreover, we use the notation $\tilde{O}(\cdot)$ to hide polynomial logarithmic factors of the problem size. As an example, $O(\log(n))$ can be expressed as $\tilde{O}(1)$.

For a matrix $\boldsymbol{\Pi} \in \mathbb{R}^{s \times n}$, we call it a Gaussian sketch if its entries are i.i.d. $\mathcal{N}(0, 1/s)$ random variables. For a diagonal matrix $\boldsymbol{R} \in \mathbb{R}^{n \times n}$, we call it a diagonal Rademacher matrix if its entries are i.i.d. Rademacher random variables. See Appendix A for more detailed introductions.

## 4    PROBLEM SETTING

Throughout this paper, we consider a graph $\mathcal{G}$ with $n$ nodes and $m$ edges. We denote its adjacency matrix by $\boldsymbol{A} \in \mathbb{R}^{n \times n}$, degree matrix by $\boldsymbol{D} = \mathrm{diag}(\boldsymbol{A1}) \in \mathbb{R}^{n \times n}$, where $\boldsymbol{1} \in \mathbb{R}^n$ is the vector with all entries equal to 1, and Laplace matrix (or Laplacian) by $\boldsymbol{L} = \boldsymbol{D} - \boldsymbol{A} \in \mathbb{R}^{n \times n}$. We also use the normalized adjacency $\hat{\boldsymbol{A}} = \boldsymbol{D}^{-1/2}\boldsymbol{A}\boldsymbol{D}^{-1/2}$ and normalized Laplacian $\hat{\boldsymbol{L}} = \boldsymbol{I} - \hat{\boldsymbol{A}}$. Notice that we assume there's a self-loop for each node to avoid 0-degree nodes, which is commonly ensured in GNN implementations (Kipf & Welling, 2016; Hamilton et al., 2017), and therefore $\boldsymbol{D}^{-1/2}$ always exists. We also use $\boldsymbol{B} \in \mathbb{R}^{m \times n}$ to represent the incidence matrix of graph $\mathcal{G}$, that is, if the $i$-th edge in $\mathcal{G}$ is $(u_i, v_i)$, then the $i$-th row of $\boldsymbol{B}$ is $\boldsymbol{\delta}_{u_i} - \boldsymbol{\delta}_{v_i}$ (the order of $u_i$ and $v_i$ is arbitrary). It is not hard to see that $\boldsymbol{L} = \boldsymbol{B}^\top \boldsymbol{B}$.

We also assume that for each node $u$ in the graph, there is a feature vector $\boldsymbol{x}_u \in \mathbb{R}^d$, and a label vector[1] $\boldsymbol{y}_u \in \mathbb{R}^c$ attached with it. Let $\boldsymbol{X} \in \mathbb{R}^{n \times d}$ and $\boldsymbol{Y} \in \mathbb{R}^{n \times c}$ be the stack of $\boldsymbol{x}_i$-s and $\boldsymbol{y}_i$-s. We assume $n \geq \max\{d, c\}$ and $\boldsymbol{X}$ and $\boldsymbol{Y}$ are both column full rank.

The Unfolded GNN we consider in this paper is based on TWIRLS (Yang et al., 2021b), which is defined by optimizing the following objective (called the "energy function"):

$$E(\boldsymbol{Z}; \boldsymbol{W}) := \frac{\lambda}{2} \operatorname{Tr}\left(\boldsymbol{Z}^\top \hat{\boldsymbol{L}} \boldsymbol{Z}\right) + \frac{1}{2} \|\boldsymbol{Z} - f(\boldsymbol{X}; \boldsymbol{W})\|_{\mathcal{F}}^2. \tag{4.1}$$

Here $\boldsymbol{Z} \in \mathbb{R}^{n \times c}$ is the optimization variable that can be viewed as the hidden state of the model, $\boldsymbol{W}$ is the learnable parameter of the model, and $\lambda > 0$ is a hyper-parameter to balance the graph structure information and node feature information. Note that $f : \mathbb{R}^d \to \mathbb{R}^c$ is a function of $\boldsymbol{X}$ and is parameterized by matrix $\boldsymbol{W}$.

For the model training, given a loss function $\ell : \mathbb{R}^{n \times c} \times \mathbb{R}^{n \times c} \to \mathbb{R}$, the training of TWIRLS can be represented by the following bi-level optimization problem

$$\boldsymbol{W}^* \in \arg\min_{\boldsymbol{W} \in \mathbb{R}^{d \times c}} \ell\left[\boldsymbol{Z}^*(\boldsymbol{W}); \boldsymbol{Y}\right] \tag{4.2}$$

$$\text{s.t.} \quad \boldsymbol{Z}^*(\boldsymbol{W}) \in \arg\min_{\boldsymbol{Z} \in \mathbb{R}^{n \times c}} E(\boldsymbol{Z}; \boldsymbol{W}), \tag{4.3}$$

where the solution of inner problem eq.(4.3) (i.e. $\boldsymbol{Z}^*(\boldsymbol{W})$) is viewed as the model output, and the outer problem eq.(4.2) is viewed as the problem of training.

In (Yang et al., 2021b), the forward process of TWIRLS is obtained by unfolding the gradient descent algorithm for minimizing $E$ given a fixed $\boldsymbol{W}$:

$$\boldsymbol{Z}^{(t+1)} = \boldsymbol{Z}^{(t)} - \alpha \nabla E\left(\boldsymbol{Z}^{(t)}\right) \tag{4.4}$$

$$= (1 - \alpha\lambda - \alpha)\boldsymbol{Z}^{(t)} + \alpha \hat{\boldsymbol{A}} \boldsymbol{Z}^{(t)} + \alpha f(\boldsymbol{X}; \boldsymbol{W}), \tag{4.5}$$

where $\alpha > 0$ is the step size of gradient descent. Notice that we fix $\boldsymbol{W}$ and view $E$ as a function of $\boldsymbol{Z}$. The model output is $\boldsymbol{Z}^{(T)}$ for a fixed iteration number $T$. When $\alpha$ is set to a suitable value and $T$ is large, it is clear that $\boldsymbol{Z}^{(T)}$ is an approximation of $\boldsymbol{Z}^*$ defined in eq.(4.3).

When $T \to \infty$, the output of TWIRLS converges to the unique global optimum of eq.(4.1), which has a closed form

$$\boldsymbol{Z}^*(\boldsymbol{W}) = \arg\min_{\boldsymbol{Z}} E(\boldsymbol{Z}) = \left(\boldsymbol{I} + \lambda \hat{\boldsymbol{L}}\right)^{-1} f(\boldsymbol{X}; \boldsymbol{W}). \tag{4.6}$$

In this paper, we always assume $T \to \infty$ and thus we focus on the closed-form solution defined by eq.(4.6).

The training problem is then defined as optimizing $\boldsymbol{W}$ to minimize the loss function between the model output $\boldsymbol{Z}^*(\boldsymbol{W})$ and label $\boldsymbol{Y}$, i.e. $\ell : \mathbb{R}^{n \times d} \times \mathbb{R}^{n \times c} \to \mathbb{R}$.

## 5 ALGORITHM AND ANALYSIS

In this section we present our main algorithm HERTA, give the convergence result and analyze its time complexity. We also introduce our key techniques and give a proof sketch for the main result. To start we introduce the following notion.

**Definition 5.1** (Effective Laplacian dimension)**.** *For normalized Laplacian $\hat{\boldsymbol{L}}$ and regularization term $\lambda > 0$, define $n_\lambda := \operatorname{Tr}\left[\hat{\boldsymbol{L}}\left(\hat{\boldsymbol{L}} + \lambda^{-1}\boldsymbol{I}\right)^{-1}\right]$ to be the effective Laplacian dimension of the graph.*

Denote $\{\lambda_i\}_{i=1}^n$ as the eigenvalues of $\hat{\boldsymbol{L}}$, then the effective Laplacian dimension can be written as $n_\lambda = \sum_i \frac{\lambda_i}{\lambda_i + \lambda^{-1}}$. We can see that $n_\lambda$ is roughly the number of eigenvalues of $\hat{\boldsymbol{L}}$ which are of order

---

[1] We allow the label to be a vector to make the concept more general. When labels are one-hot vectors, the task is a classification task, and when labels are scalars, it is a regression task.

$\Omega(\lambda^{-1})$, i.e., the number of "large eigenvalues" mentioned in Theorem 1.1. It is not hard to see that $n_\lambda \le n$ and $n_\lambda \to n$ as $\lambda \to \infty$.

Intuitively, $n_\lambda$ represents the number of eigen-directions for which the Laplacian regularizer $\frac{\lambda}{2}\mathrm{Tr}(\boldsymbol{Z}^\top \hat{\boldsymbol{L}} \boldsymbol{Z})$ is large, thus having a significant impact on the energy function equation 4.1. For the remaining eigen-directions which are less significant, the effect of the regularizer is minimal. This can also be seen from the closed form solution of the model equation 4.6: when we apply matrix $(\boldsymbol{I} + \lambda \hat{\boldsymbol{L}})^{-1}$ to a vector, the effect of the Laplacian is dominated by the $\boldsymbol{I}$ term if this vector is aligned with a "small" Laplacian eigen-direction (i.e., with an eigenvalue significantly smaller than $\lambda^{-1}$), and thus the objective is close to the usual unregularized least squares loss.

Based on the above discussion, it stands to reason that the larger the effective Laplacian dimension $n_\lambda$ of the graph, the more work we must do during the training to preserve the graph structure of the GNN.

## 5.1 MAIN RESULT

In this subsection, we present our main theoretical result. Before delving into the specific theorem, we first outline the model implementation used in our theoretical analysis. Although we make specific assumptions for this analysis, the experiments (see Section 6) demonstrate that our proposed algorithm is effective in broader settings beyond those considered here.

Principally, the implementation of $f(\boldsymbol{X}; \boldsymbol{W})$ can be arbitrary as long as it can be trained (i.e. computable, smooth, etc.). However, we notice that in Yang et al. (2021b), most of the SOTA results are achieved by the implementation $f(\boldsymbol{X}; \boldsymbol{W}) = \boldsymbol{X}\boldsymbol{W}$ where $\boldsymbol{W} \in \mathbb{R}^{d \times c}$. Therefore, hereinafter we focus on this specific implementation.

For the simplicity of analysis, we present our theoretical result based on the mean squared error (MSE) loss defined as $\ell(\boldsymbol{Z}, \boldsymbol{Y}) := \frac{1}{2}\|\boldsymbol{Z} - \boldsymbol{Y}\|_{\mathcal{F}}^2$. Despite this specific choice in the analysis, we empirically show in Section 6 that our algorithm also works for cross entropy (CE) loss, which is the most commonly used loss function for node classification tasks.

Under MSE loss, we can decompose the loss into sub-losses for each class, i.e. $\ell[\boldsymbol{Z}^*(\boldsymbol{W}); \boldsymbol{Y}] = \sum_{i=1}^c \ell_i(\boldsymbol{w}_i; \boldsymbol{y}_i)$, where $\boldsymbol{w}_i$ and $\boldsymbol{y}_i$ are the $i$-th column of $\boldsymbol{W}$ and $\boldsymbol{Y}$ respectively. In the following analysis, we will mostly focus on each $\ell_i$. Moreover, we fix $\boldsymbol{y}_i$, and view $\ell_i$ as a function of $\boldsymbol{w}_i$.

**Theorem 5.1** (Main result). *For any $\epsilon > 0$, with a proper step size $\eta$, number of iterations $T$ and constant $K > 0$, HERTA (Algorithm 1) finds a solution $\hat{\boldsymbol{W}} \in \mathbb{R}^{d \times c}$ in time*

$$\tilde{O}\left((m + nd)\, c \left(\log \frac{1}{\epsilon}\right)^2 + n_\lambda \lambda^2 d + d^3\right) \tag{5.1}$$

*such that with probability $1 - \frac{1}{n}$, $\ell\left[\boldsymbol{Z}^*(\hat{\boldsymbol{W}}); \boldsymbol{Y}\right] \le (1 + \epsilon) \cdot \ell^*$, where $\ell^* := \min_{\boldsymbol{W}} \ell\left[\boldsymbol{Z}^*(\boldsymbol{W}; \boldsymbol{Y})\right]$.*

Notice that if we assume $n_\lambda = O(n/\lambda^2)$, then $\tilde{O}(n_\lambda \lambda^2 d) = \tilde{O}(nd)$. By further assuming the number of classes $c = O(1)$, we recover the bound given by Theorem 1.1.

**Technical contributions.** Before heading into the details of HERTA (Algorithm 1), we present our two main technical contributions that go into the proof of Theorem 5.1.

1. Our first technical contribution is a rigorous convergence and complexity analysis of solving the bilevel optimization problem given by eq.(4.6). Crucially, our analysis allows the inner problem to be solved inexactly, thus enabling integration with Laplacian system solvers.

2. Our second main technical contribution is a new spectral sparsification algorithm, which improves on the state-of-the-art approaches by leveraging the regularized and normalized graph Laplacians that arise in this task.

## 5.2 CONVERGENCE ANALYSIS WITH INEXACT SUB-PROBLEMS

In this subsection, we develop a general-purpose convergence analysis framework, which captures not only our algorithms but also some prior approaches. We then use this framework to provide a

complexity analysis of the training algorithm of Yang et al. (2021b), pointing out the key bottlenecks in their approach that we address with HERTA.

As noted in Section 4, the whole optimization problem can be viewed as a bi-level optimization problem, where the inner-problem eq.(4.3) approximates the linear system solver $\left(\boldsymbol{I}+\lambda\hat{\boldsymbol{L}}\right)^{-1}(\boldsymbol{X}\boldsymbol{w}_i - \hat{\boldsymbol{y}}_i)$, and the outer problem eq.(4.2) uses the linear system solution to approximate the optimal solution of the training loss. When we discuss the solution of outer problems, we treat the solver of the inner problem (linear solver) as a black box. We formally define a linear solver as follows.

**Definition 5.2** (Linear solver). *For a positive definite matrix $\boldsymbol{H} \in \mathbb{R}^{n\times n}$ and a real number $\epsilon > 0$, we call $f : \mathbb{R}^n \to \mathbb{R}^n$ a linear solver for $\boldsymbol{H}$ with $\epsilon$ error rate if it satisfies*

$$\forall \boldsymbol{u} \in \mathbb{R}^n, \ \left\|f(\boldsymbol{u}) - \boldsymbol{H}^{-1}\boldsymbol{u}\right\|_{\boldsymbol{H}} \leq \epsilon \left\|\boldsymbol{H}^{-1}\boldsymbol{u}\right\|_{\boldsymbol{H}}. \tag{5.2}$$

We now consider solving the outer problem by gradient descent (in Section 5.4, we further augment this by preconditioning). The gradient of $\ell_i(\boldsymbol{w}_i)$ is

$$\nabla\ell_i(\boldsymbol{w}_i) = \boldsymbol{X}^{\top}\left(\boldsymbol{I}+\lambda\hat{\boldsymbol{L}}\right)^{-2}\left(\boldsymbol{X}\boldsymbol{w}_i - \hat{\boldsymbol{y}}_i\right), \tag{5.3}$$

where $\hat{\boldsymbol{y}}_i = (\boldsymbol{I}+\lambda\hat{\boldsymbol{L}})\boldsymbol{y}_i$.

Instead of computing this gradient exactly, we allow the above matrix inverse to be calculated by linear solvers. Suppose that we have a linear solver for $(\boldsymbol{I}+\lambda\hat{\boldsymbol{L}})$ denoted by $\mathcal{S}$, then by embedding it into eq.(5.3) we can obtain the gradient approximation as

$$\widetilde{\nabla\ell_i(\boldsymbol{w}_i)} := \boldsymbol{X}^{\top}\mathcal{S}\left[\mathcal{S}\left(\boldsymbol{X}\boldsymbol{w}_i - \hat{\boldsymbol{y}}_i\right)\right]. \tag{5.4}$$

The following convergence result can be derived for this approximate gradient descent. See Appendix for the proof.

**Lemma 5.1** (Convergence). *If we minimize $\ell_i$ using gradient descent with the gradient approximation defined in eq.(5.4), where $\mathcal{S}$ is a linear solver of $\boldsymbol{I}+\lambda\hat{\boldsymbol{L}}$ with $\mu$ error rate satisfying $\mu \leq \min\left\{\frac{\epsilon^{1/2}}{50\kappa(\boldsymbol{X})\lambda^2}, 1\right\}$, then, to obtain an $\epsilon \in (0,1)$ error rate the outer problem (i.e. $\ell_i$), the number of iterations needed is $T = O\left(\kappa\log\frac{1}{\epsilon}\right)$, where $\kappa := \kappa(\boldsymbol{X}^{\top}(\boldsymbol{I}+\lambda\hat{\boldsymbol{L}})^{-2}\boldsymbol{X})$.*

Based on Lemma 5.1, in Appendix B we provide a detailed analysis of the time complexity of the implementation used by Yang et al. (2021b). From this analysis, one can figure out two bottlenecks of the computational complexity for solving TWIRLS: 1) The number of iterations needed for the inner loop is dependent on $\lambda$; 2) The number of iterations needed for the outer loop is dependent on the condition number of the outer problem. Our acceleration algorithm is based on the following two observations, which correspond to the aforementioned bottlenecks:

1. The data matrix in the inner loop is a graph Laplacian plus identity, and thus, we can solve the linear system defined by this matrix faster by exploiting its structure.

2. A strong preconditioner can be efficiently constructed for the outer iteration. We do this by combining our regularized spectral sparsifier with fast approximate matrix multiplication.

## 5.3 REGULARIZED SPECTRAL SPARSIFIER

It has been shown that for any connected graph with $n$ nodes, (no matter how dense it is) there is always a sparsified graph with $\tilde{O}\left(\frac{n}{\epsilon^2}\right)$ edges whose Laplacian is an $\epsilon$-spectral approximation of the original graph Laplacian Spielman & Teng (2011). Although the existence is guaranteed, it is generally hard to construct such a sparsified graph. An algorithm that finds such a sparsified graph is called a spectral sparsifier. While there has been many existing explorations on constructing a spectral sparsifier of a given graph with high probability and tolerable time complexity Spielman & Srivastava (2008); Spielman & Teng (2011); Koutis et al. (2016), the setting considered in this paper is different from the standard one: instead of a graph Laplacian $\boldsymbol{L}$, the matrix we encounter is the normalized and regularized Laplacian $\hat{\boldsymbol{L}} + \lambda^{-1}\boldsymbol{I}$.

At first glance, a spectral sparsifier also works for our problem since $\tilde{L} \approx_\epsilon \hat{L}$ implies $\tilde{L} + \lambda^{-1}I \approx_\epsilon \hat{L} + \lambda^{-1}I$. However, by leveraging the regularization term $\lambda^{-1}I$, we can substantially reduce the associated computational cost. To this end, we propose a new spectral sparsifier that works for $\hat{L} + \lambda^{-1}I$ which is used in our problem. This spectral sparsifier reduces the number of edges to $O\left(\frac{n_\lambda}{\epsilon^2} \log n\right)$, which, as we discussed before, is smaller than $O\left(\frac{n}{\epsilon^2} \log n\right)$ obtained by standard spectral sparsifier.

---

**Algorithm 1** HERTA: A High-Efficiency and Rigorous Training Algorithm for TWIRLS

---

**Input:** $\hat{L}, X, Y, \lambda, c, K > 0$, step size $\eta$ and number of iteration $T$.

Set $\beta = \frac{1}{64}$, $s = \frac{Kd}{\beta^2} \log n$, $\mu = \min\left\{\frac{\epsilon^{1/2}}{50\kappa(X)\lambda^2}, 1\right\}$ and $R \in \mathbb{R}^{n \times n}$ a diagonal Rademacher matrix;

$\tilde{L} \leftarrow \mathsf{Sparsify}_{\frac{\beta}{3\lambda}}(\hat{L})$ by calling Algorithm 2;

$Q \leftarrow \mathsf{SSolve}_{\frac{\beta}{\sqrt{3\lambda}}}(I + \lambda\tilde{L}; X)$;

$Q' \leftarrow \mathsf{Hadamard}(RQ)$;

Subsample $s$ rows of $Q'$ uniformly and obtain $\tilde{Q}$;

$P \leftarrow \tilde{Q}^\top \tilde{Q}$;

$P' \leftarrow P^{-1/2}$;

**for** $i = 1$ **to** $c$ **do**

  $\hat{y}_i \leftarrow (I + \lambda\hat{L})y_i$, where $y_i$ is the $i$-th column of $Y$;

  Initialize $w_i^{(0)}$ by all zeros;

  **for** $t = 1$ **to** $T$ **do**

    $u_i^{(t)} \leftarrow XP'w_i^{(t-1)} - \hat{y}_i$;

    $u_i^{(t)'} \leftarrow \mathsf{SSolve}_\mu\left(I + \lambda\hat{L}; u_i^{(t)}\right)$;

    $u_i^{(t)''} \leftarrow \mathsf{SSolve}_\mu\left(I + \lambda\hat{L}; u_i^{(t)'}\right)$;

    $g_i^{(t)} \leftarrow P'X^\top u_i^{(t)''}$;

    $w_i^{(t)} \leftarrow w_i^{(t-1)} - \eta g_i^{(t)}$;

  **end for**

**end for**

**Output:** $\left\{w_i^{(T)}\right\}_{i=1}^c$.

---

**Algorithm 2** $\mathsf{Sparsify}_\epsilon(\hat{L})$: Regularized Spectral Sparsifier

---

**Input:** $\hat{L}, \hat{B}, \lambda, C > 0$ and expected error rate $\epsilon$.

Set $k = C \log m$ and $s = \frac{Cn_\lambda \log n}{\epsilon^2}$, and construct Gaussian sketch $\Pi_1 \in \mathbb{R}^{k \times m}, \Pi_2 \in \mathbb{R}^{k \times n}$;

$B_{\mathcal{S}} \leftarrow \mathsf{SSolve}_{2^{\frac{1}{4}}}(\hat{L} + \frac{1}{\lambda}I; (\Pi_1\hat{B})^\top)$;

$\Pi_{\mathcal{S}} \leftarrow \mathsf{SSolve}_{2^{\frac{1}{4}}}(\hat{L} + \frac{1}{\lambda}I; \Pi_2^\top)$;

**for** $i = 1$ **to** $m$ **do**

  $\tilde{l}_i \leftarrow \|B_{\mathcal{S}}\hat{b}_i\|^2 + \lambda^{-1}\|\Pi_{\mathcal{S}}\hat{b}_i\|^2$,

  where $\hat{b}_i^\top$ is the $i$-th row of $\hat{B}$;

**end for**

$Z \leftarrow \sum_{i=1}^m \tilde{l}_i$;

Subsample $s$ rows of $\hat{B}$ with probabilities $\left\{\tilde{l}_i/Z\right\}_{i=1}^m$ and obtain $\tilde{B}$;

$\tilde{L} \leftarrow \tilde{B}^\top \tilde{B}$;

**Output:** $\tilde{L}$.

---

**Lemma 5.2** (Regularized spectral sparsifier)**.** *Let $\hat{L}$ be a (normalized) graph Laplacian with $m$ edges and $n$ nodes. There exists a constant $C$ such that for any $\epsilon > 0$ and $\lambda > 0$, Algorithm 2 outputs a (normalized) graph Laplacian $\tilde{L}$ with $n$ nodes and $O\left(\frac{n_\lambda}{\epsilon^2} \log n\right)$ edges in time $\tilde{O}(m)$, such that $\tilde{L} + \lambda^{-1}I \approx_\epsilon \hat{L} + \lambda^{-1}I$ with probability at least $1 - \frac{1}{2n}$.*

To achieve this result, we modify the sub-sampling probabilities needed for selecting the edges into the sparsified graph, leveraging the regularization term $\lambda^{-1}I$ via the notion of ridge leverage scores Cohen et al. (2017b). Another technique we use in Algorithm 2 is a symmetric diagonal dominated (SDD) system solver. Specifically, if a symmetric matrix $H \in \mathbb{R}^{n \times n}$ satisfies

$$\forall 1 \le k \le n, \ \ H_{k,k} \ge \sum_{i=1}^n |H_{k,i}|, \tag{5.5}$$

it is called diagonal dominated. It has been shown that the linear system defined by sparse SDD matrices can be solved in an efficient way, via a SDD solver Kelner et al. (2013); Peng & Spielman (2014). Since $I + \lambda\hat{L}$ is indeed a SDD matrix in our problem, we can apply off-the-shelf SDD solvers. In the following, we use $\mathsf{SSolve}_\epsilon(H, u)$ to denote the SDD solver that calculates $H^{-1}u$ with error

rate less or equal to $\epsilon^2$. From Lemma A.1, the time complexity of calculating $\mathsf{SSolve}_\epsilon(\boldsymbol{H}; \boldsymbol{u})$ is $\tilde{O}\left[\mathrm{nnz}(\boldsymbol{H})\right]$. See Appendix A.2 for details.

## 5.4 MAIN ALGORITHM

Now we are ready to present our main algorithm HERTA, see Algorithm 1. As mentioned before, HERTA is composed of two major components: constructing a preconditioner matrix $\boldsymbol{P}$ and applying it to the optimization problem. Below we introduce each part.

**Constructing the Preconditioner.** For the outer problem which can be ill-conditioned, we first precondition it using a preconditioner $\boldsymbol{P} \in \mathbb{R}^{d \times d}$ which is a constant level spectral approximation of the Hessian, i.e. $\boldsymbol{X}^\top \left(\boldsymbol{I} + \lambda \hat{\boldsymbol{L}}\right)^{-2} \boldsymbol{X}$. We claim that the matrix $\boldsymbol{P}$ constructed inside Algorithm 1 indeed satisfies this property.

**Lemma 5.3** (Preconditioner). *Let $\boldsymbol{P}$ be the matrix constructed in Algorithm 1. We have $\boldsymbol{P} \approx_{\frac{1}{2}}$ $\boldsymbol{X}^\top \left(\boldsymbol{I} + \lambda \hat{\boldsymbol{L}}\right)^{-2} \boldsymbol{X}$ with probability at least $1 - \frac{1}{n}$.*

Notice that when constructing the preconditioner $\boldsymbol{P}$ in Algorithm 1, one critical step is to calculate $\boldsymbol{Q}^\top \boldsymbol{Q}$, where $\boldsymbol{Q} \in \mathbb{R}^{n \times d}$ where $n > d$. This matrix multiplication is known to take $O(nd^2)$ time if implemented by brute force. Therefore, in Algorithm 1, we use Subsampled Randomized Hadamard Transformat (SRHT) Tropp (2011); Woodruff et al. (2014) to achieve fast matrix multiplications. The key idea of SRHT is to first apply a randomized Hadamard transformation to the matrix to make the "information" evenly distributed in each row, then uniformly sample the rows of $\boldsymbol{Q}$ to reduce the dimension. In Algorithm 1, we use Hadamard to represent the Hadamard transformation, see Appendix A.3 for the definition of Hadamard as well as a detailed introduction.

**Solving the Outer Problem.** After obtaining the preconditioner $\boldsymbol{P}$ which approximates the Hessian by a constant level, we use it to precondition the outer problem and provably reduce the condition number to a constant. With the new problem being well-conditioned, iterative methods (like gradient descent) take much less steps to converge.

**Lemma 5.4** (Well-conditioned Hessian). *Let $\ell'$ be such that $\ell'(\boldsymbol{w}_i) = \ell(\boldsymbol{P}^{-1/2}\boldsymbol{w}_i)$. Suppose for some constant $c_0 \in (0, 1)$ we have $\boldsymbol{P} \approx_{c_0} \boldsymbol{X}^\top \left(\boldsymbol{I} + \lambda \hat{\boldsymbol{L}}\right)^{-2} \boldsymbol{X}$, then the condition number of the Hessian of $\ell'$ is bounded by*

$$\kappa\left(\nabla^2 \ell'\right) \leq (1 + c_0)^2. \tag{5.6}$$

*Moreover, if $\boldsymbol{w}'$ is a solution of $\ell'$ with $\epsilon$ error rate, then $\boldsymbol{P}^{-1/2}\boldsymbol{w}'$ is a solution of $\ell$ with $\epsilon$ error rate.*

Notice that, the problem $\ell'$ defined in Lemma 5.4 can be viewed as the original problem $\ell$ with $\boldsymbol{X}$ being replaced by $\boldsymbol{X}\boldsymbol{P}^{-1/2}$, therefore we can use Lemma 5.1 and obtain the convergence rate of HERTA. With the convergence result and analysis of the running time for each step, we are able to prove Theorem 5.1. See Appendix C.7 for the full proof.

**Remark on the Unavoidable $\lambda^2$ in Runtime.** Notice that $\lambda^2$ appears in the time bound given in Theorem 5.1. From the proof in Appendix C.5 we can see that this term originates from the squared inverse of $\boldsymbol{I} + \lambda \hat{\boldsymbol{L}}$. In Appendix D.2, we show that in the worst case even if we approximate a matrix to a very high precision, the approximation rate of the corresponding squared version can still be worse by a factor of the condition number of the matrix, which in our case leads to the unavoidable $\lambda^2$ in the runtime.

**Remark on Sparsifying the Graph in Each Iteration.** Note that in Algorithm 1, we use the complete Laplacian $\hat{\boldsymbol{L}}$ for gradient iterations (i.e., the for-loop), and the graph sampling only occurs when constructing the preconditioner $\boldsymbol{P}$. We note that sampling $\hat{\boldsymbol{L}}$ in the for-loop can lead to extra loss in the gradient estimation, which forces us to sparsify the graph to a very high precision to

---

[2]Principally, $\mathsf{SSolve}(\boldsymbol{H}; \cdot)$ is a vector function, but for convenience we sometimes also apply it to matrices, in which case we mean applying it column-wisely. We also use the same convention for other vector functions.

ensure an accurate enough gradient for convergence. This could result in a suboptimal running time. Moreover, as discussed in Section 1, the current running time bound for HERTA is optimal up to logarithmic factors, which means that there is little to gain from performing extra sampling in for-loops. See Appendix D.3 for a more detailed and quantitative discussion. The experimental results show that HERTA significantly outperforms standard optimizers, such as gradient descent and Adam in reducing both training and evaluation errors across different datasets and configurations.

## 6 EXPERIMENTS

In this section, we verify our theoretical results through experiments on five real world datasets. Since for the inner problem we use off-the-shelf SDD solvers, in Sections 6.1 and 6.2 we focus on the outer problem, i.e. the training loss convergence rate. In each setting we compare the training loss convergence rate of TWIRLS trained by our method against that trained by standard optimizers using exactly the same training hyper-parameters. Moreover, in Section 6.3 we compare the generalization performance of TWIRLS trained with HERTA and standard optimizers.

**Datasets.** We conduct experiments on five datasets, including all of the ones used in the Section 5.1 of Yang et al. (2021b): Cora, Citeseer and Pubmed collected by Sen et al. (2008), as well as the ogbn-arxiv dataset from the Open Large Graph Benchmark (OGB, Hu et al., 2020). Moreover, we also evaluate on Flickr, which was introduced by Zeng et al. (2019).

Notice that for Cora, Citeseer and Pubmed, common practice uses the semi-supervised setting where there is a small training set and relatively large validation set. As we are comparing training convergence rate, we find it more comparative to use a larger training set (which makes solving the optimization problem of training more difficult). Therefore, we randomly select 80% of nodes as training set for Cora, Citeseer and Pubmed. For OGB, we use the standard training split.

Due to limited space, we only include the results with larger datasets in this section, and for comparison of training loss, we only present the results with $\lambda = 1$. Additional results, including on the other datasets and with more hyper-parameter values, are deferred to Appendix F.

### 6.1 CONVERGENCE RATE COMPARISON UNDER MSE LOSS

In this section, we compare the convergence rate for models trained with MSE loss, which is well aligned with the setting used to derive our theory. We also adopt a variation of HERTA that allows using it on other optimizers and apply it on Adam optimizer. See Figure 1 for the results. Notice that for each figure, we shift the curve by the minimum value of the loss (which is approximated by running HERTA for more epochs) and take logarithm scale to make the comparison clearer.

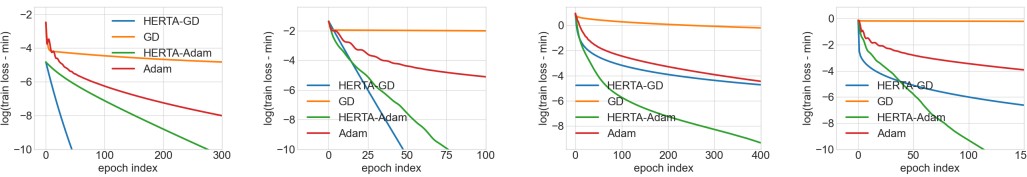

Figure 1: The training loss comparison between HERTA and standard optimizers on MSE loss with $\lambda = 1$. Left: ogbn-arxiv; Right: Pubmed.

Figure 2: The training loss comparison between HERTA and standard optimizers on CE loss with $\lambda = 1$. Left: ogbn-arxiv; Right: Pubmed.

From the results, it is clear that on all datasets and all optimizers we consider, HERTA converges much faster than standard methods. It generally requires $\leq 10$ iterations to get very close to the smallest training loss. This shows that the guarantee obtained in our theoretical results (Theorem 5.1) not only matches the experiments, but also holds when the setting is slightly changed (using other optimizers). Thus, these experimental results verify the universality of HERTA as an optimization framework.

We also conduct extensive experiments with larger $\lambda$. See Appendix F for the results. These results verify that even when $\lambda$ is relatively large, HERTA converges very fast, which suggests that the dominant term in Theorem 5.1 is the $\tilde{O}(m)$ term instead of the $\tilde{O}(n_\lambda \lambda^2 d)$ term.

## 6.2 Convergence Rate Comparison under Cross Entropy Loss

Note that in the theoretical analysis of Section 5, we focus on the MSE loss. However, for graph node classification tasks, MSE is not the most commonly used loss. Instead, most node classification models use the cross entropy (CE) loss, e.g., see Yang et al. (2021b).

To this end, we also adopt HERTA on training problems with CE loss. See Appendix E for the details of the implementation. The results are displayed in Figure 2. From these results, it is clear that HERTA also significantly speeds up training with CE loss. The results demonstrate that although originally developed on MSE loss, HERTA is not limited to it and has the flexibility to be applied to other types of loss functions.

**Remark on the Surprising Effectiveness of HERTA on Cross Entropy Loss.** We claim that this phenomenon might originate from the fact the Hessians of TWIRLS under MSE loss and CE loss are very similar. We provide an analysis of the gradient and Hessian of TWIRLS under MSE and CE losses in Appendix D.1. The results show that the gradient under CE loss can be viewed as the gradient under MSE loss with one term being normalized by softmax, and the Hessian under CE loss can be viewed as a rescaled version of the Hessian under MSE loss. These comparisons serve as an explanation of why HERTA works so well with CE loss.

## 6.3 Test Accuracy Comparison

Despite the fact that the main focus of HERTA is accelerating the training, here we also show that HERTA has excellent generalization properties, by comparing the accuracy of TWIRLS trained by HERTA and standard optimizers on a held out test set.

To obtain a fair test accuracy comparison, for each dataset and algorithm we do a grid search of the hyper-parameters including learning rate, input dropout rate and $\lambda$. See Appendix E for the details of the grid search. We select the results based on the optimal validation set accuracy. Due to limited space, here we present in Figure 3 the test accuracy curve for TWIRLS trained with MSE loss on ogbn-arxiv, Pubmed and Flickr and defer other results to Appendix F.

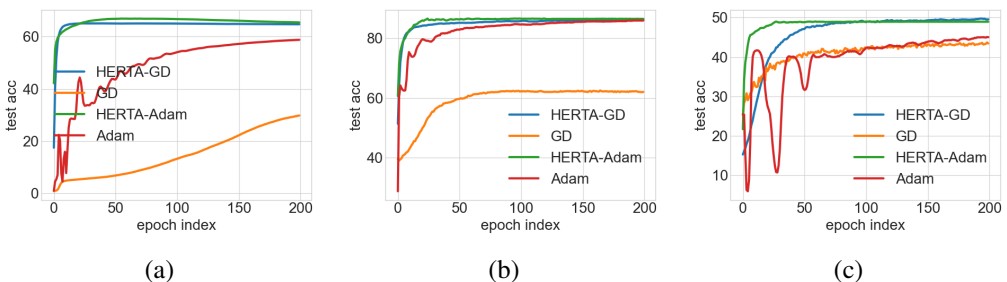

(a)           (b)           (c)

Figure 3: The test accuracy comparison between HERTA and standard optimizers trained with MSE loss. (a) ogbn-arxiv; (b) Pubmed; (c) Flickr.

## 7 Conclusion

In this paper we present HERTA: a High-Efficiency and Rigorous Training Algorithm that solves the problem of training Unfolded GNNs on a graph within a time essentially of the same magnitude as the time it takes to load the input data. As a component of HERTA, we also propose a new spectral sparsifier that works for normalized and regularized graph Laplacian matrices.

Experimental results on real world datasets show the effectiveness of HERTA. Moreover, it is shown that HERTA works for various loss functions and optimizers, despite being derived from a specific loss function and optimizer. Moreover, while being derived from training perspective, HERTA is also capable of accelerating generalization. These results highlight the universality of HERTA and confirm that it is ready to use in practice. See Appendix D for further discussions and future directions.

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

# A INTRODUCTION TO THE MATHEMATICAL TOOLS

In the main paper, due to space limitations, our introduction of certain mathematical tools and concepts remains brief. In this section, we provide a more comprehensive introduction to the mathematical tools utilized in the main paper.

## A.1 SUBSAMPLING

Subsampling is used in multiple places in our Algorithm. When we say subsampling $s$ rows of a matrix $\boldsymbol{M} \in \mathbb{R}^{n \times d}$ with probability $\{p_k\}_{k=1}^n$ and obtain a new matrix $\tilde{\boldsymbol{M}} \in \mathbb{R}^{s \times d}$, it means each row of $\tilde{\boldsymbol{M}}$ is an i.i.d. random vector which follows the same distribution of a random vector $\boldsymbol{\xi}$ that satisfies the following property:

$$\mathbb{P}\left\{\boldsymbol{\xi} = \sqrt{\frac{1}{sp_k}}\boldsymbol{m}_k\right\} = p_k, \tag{A.1}$$

where $\boldsymbol{m}_k$ is the $k$-th row of matrix $\boldsymbol{M}$. It is clear that constructing such a subsampled matrix $\tilde{\boldsymbol{M}}$ takes time $O(sd)$. Moreover, if $p_k = \frac{1}{n}$, we say the subsampling is with uniform probability.

The operator of subsampling $s$ rows of a $n$-row matrix is clearly a linear transformation. Therefore it can be represented by a random matrix $\boldsymbol{S} \in \mathbb{R}^{s \times n}$. Notice that when we apply $\boldsymbol{S}$ to a matrix (i.e. $\boldsymbol{S} \cdot \boldsymbol{M}$) we don't actually need to construct such a matrix $\boldsymbol{S}$ and calculate matrix product, since we only carry out sampling.

## A.2 SDD SOLVERS

As mentioned in the main paper, for a sparse SDD matrix $\boldsymbol{M}$, it is possible to fast approximate its linear solver. The following Lemma A.1 rigorously states the bound we can obtain for SDD solvers.

**Lemma A.1** (Peng & Spielman (2014)). *For any SDD matrix $\boldsymbol{M} \in \mathbb{R}^{n \times n}$ and any real number $\epsilon > 0$, there exists an algorithm to construct an operator $\mathsf{SSolve}_\epsilon(\boldsymbol{M}; \cdot)$, such that $\mathsf{SSolve}_\epsilon(\boldsymbol{M}; \cdot)$ is a linear solver for $\boldsymbol{M}$ with $\epsilon$ error rate and for any $\boldsymbol{x} \in \mathbb{R}^n$, calculating $\mathsf{SSolve}_\epsilon(\boldsymbol{M}; \boldsymbol{x})$ takes time*

$$O\left(m\left(\log\frac{1}{\epsilon}\right)\operatorname{poly}\log(m)\operatorname{poly}\log(\kappa(\boldsymbol{M}))\right), \tag{A.2}$$

*where $m = \operatorname{nnz}(\boldsymbol{M})$.*

## A.3 FAST MATRIX MULTIPLICATION

As mentioned in the main paper, we use SRHT to achieve fast matrix multiplication. Here we further explain this process. We refer interested readers to Tropp (2011) for more details. For a positive integer number $n$ that is a power of 2, the Hadamard transformation of size $n$ is a linear transformation recursively defined as follows:

$$\boldsymbol{H}_n := \begin{bmatrix} \boldsymbol{H}_{n/2} & \boldsymbol{H}_{n/2} \\ \boldsymbol{H}_{n/2} & -\boldsymbol{H}_{n/2} \end{bmatrix} \tag{A.3}$$

and $\boldsymbol{H}_1 = 1$. For a vector $\boldsymbol{x} \in \mathbb{R}^n$, we define

$$\mathsf{Hadamard}(\boldsymbol{x}) := \frac{1}{\sqrt{n}}\boldsymbol{H}_n\boldsymbol{x}. \tag{A.4}$$

Notice that when $n$ is not a power of 2, we pad the vector $\boldsymbol{x}$ with 0-s beforehand, therefore the requirement of $n$ being a power of 2 is ignored in the actual usage. From its recursive nature, it is not hard to see that applying a Hadamard transformation to each column of an $n \times d$ matrix only takes $O(d \log n)$ time.

Let $\boldsymbol{R} \in \mathbb{R}^{n \times n}$ be a diagonal matrix whose diagonal entries are i.i.d. Rademacher variables, and $\boldsymbol{S} \in \mathbb{R}^{s \times n}$ be a subsampling matrix with uniform probability and $s = O\left(\frac{d}{\beta^2}\log n\right)$. For a matrix $\boldsymbol{Q} \in \mathbb{R}^{n \times d}$, its SRHT with $\beta$ error rate is defined as

$$\mathsf{SRHT}_\beta(\boldsymbol{Q}) := \boldsymbol{S}\mathsf{Hadamard}(\boldsymbol{R}\boldsymbol{Q}), \tag{A.5}$$

which is exactly the matrix $\tilde{Q}$ we use in Algorithm 1. A fast matrix multiplication result can be achieved using SRHT.

**Lemma A.2** (Tropp (2011)). *For matrix $Q \in \mathbb{R}^{n \times d}$ where $n \geq d$ and $\text{rank}(Q) = d$, let $\tilde{Q} = \text{SRHT}_\beta(Q)$ where $\beta \in (0, 1/4)$, then we have*

$$\tilde{Q}^\top \tilde{Q} \approx_\beta Q^\top Q \tag{A.6}$$

*with probability at least $1 - \frac{1}{2n}$.*

# B   RUNNING TIME OF THE ORIGINAL IMPLEMENTATION USED IN YANG ET AL. (2021B)

In this section we analyze the time complexity of the implementation used in Yang et al. (2021b), which uses gradient descent to solve both inner and outer problem.

**Inner Problem Analysis.** We first analyze the time complexity of the inner problem solver used in Yang et al. (2021b). Recall that for the inner problem eq.(4.3), we need to find an approximation of the linear solver of $I + \lambda \hat{L}$. In Yang et al. (2021b), it is implemented by a standard gradient descent. Here we consider approximately solving the following least square problem by gradient descent:

$$v = \arg \min_{v \in \mathbb{R}^n} \frac{1}{2} \left\| \left( I + \lambda \hat{L} \right) v - u \right\|_2^2. \tag{B.1}$$

The gradient step with step size $\mu$ is:

$$v^{(t+1)} = (1 - \mu) \left( I + \lambda \hat{L} \right)^2 v^{(t)} + \eta \left( I + \lambda \hat{L} \right) u. \tag{B.2}$$

**Theorem B.1** (Inner Analysis). *If we update $v^{(t)}$ through eq.(B.2) with initialization $v^{(0)} = 0$ and a proper $\eta$, then after $T = O\left( \lambda^2 \log \frac{\lambda}{\epsilon^2} \right)$ iterations we can get*

$$\left\| v^{(T)} - \left( I + \lambda \hat{L} \right)^{-1} u \right\|_{(I + \lambda \hat{L})} \leq \epsilon \left\| \left( I + \lambda \hat{L} \right)^{-1} u \right\|_{(I + \lambda \hat{L})} \tag{B.3}$$

*for any $\epsilon \in (0, 1)$.*

*Proof.* Let $H = I + \lambda \hat{L}$. Notice that the strongly convexity and Lipschitz constant of Problem eq.(B.1) is $\mathscr{C} = 1$ and $L \leq 9\lambda^2$ respectively. From Lemma C.1, take $\eta = \frac{1}{L}$ and $v^{(0)} = 0$, we have

$$\| H v^{(T)} - u \|^2 \leq (1 - 9\lambda^2)^T \| u \|^2 \tag{B.4}$$

for any $T \in \mathbb{N}$. Therefore,

$$\frac{\| v^{(T)} - H^{-1} u \|_H^2}{\| H^{-1} u \|_H^2} = \frac{\| H v^{(T)} - u \|_{H^{-1}}^2}{\| u \|_{H^{-1}}^2} \tag{B.5}$$

$$\leq 3\lambda \frac{\| H v^{(T)} - u \|_2^2}{\| u \|_2^2} \tag{B.6}$$

$$\leq 3\lambda (1 - 9\lambda^2)^T. \tag{B.7}$$

To obtain an error rate $\epsilon$, the smallest number of iterations $T$ needed is

$$T = \left( \log \frac{1}{1 - (9\lambda^2)^{-1}} \right)^{-1} \log \frac{3\lambda}{\epsilon^2} + 1 = O\left[ \left( \log \frac{1}{1 - \lambda^{-2}} \right)^{-1} \log \frac{\kappa}{\epsilon^2} \right] = O\left( \lambda^2 \log \frac{\lambda}{\epsilon^2} \right). \tag{B.8}$$

$\square$

**Overall Running Time.** As we proved above in Theorem B.1, the number of iterations needed for solving the inner problem to error rate $\beta$ with the implementation of Yang et al. (2021b) is $\tilde{O}\left(\lambda^2 \log \frac{1}{\beta}\right)$, which is related to the hyper-parameter $\lambda$. For each inner iteration (i.e. eq.(B.2)), we need to compute $(\boldsymbol{I} + \lambda \boldsymbol{L})(\boldsymbol{X}\boldsymbol{w} - \hat{\boldsymbol{y}})$. Since this is a sparse matrix multiplication, the complexity of this step is $O(m + nd)$. Putting things together, we have the time complexity of calling inner problem solver is $\tilde{O}\left((m + nd)\lambda^2 \log \frac{1}{\beta}\right)$.

From Lemma 5.1, the number of outer iterations needed is $\tilde{O}\left(\kappa_o \log \frac{1}{\epsilon}\right)$, where $\kappa_o$ is the condition number of the outer problem. Moreover, Lemma 5.1 also indicates that we require $\beta \leq \frac{\epsilon^{1/2}}{25\kappa(\boldsymbol{X})\lambda}$, so solving inner problem takes $\tilde{O}(\lambda^2 \log 1/\beta) = \tilde{O}(\lambda^2 \log 1/\epsilon)$. In each outer iteration, we need to call inner problem solver constant times, and as well as performing constant matrix vector multiplications whose complexity is $O(nd)$. Therefore the overall running time of solving the training problem is $\tilde{O}\left[\kappa_o\left(\lambda^2(m + nd) + nd\right)\left(\log \frac{1}{\epsilon}\right)^2\right]$.

## C PROOF OF THEORETICAL RESULTS

We first note that since $\hat{\boldsymbol{L}}$ is normalized, all the eigenvalues of $\hat{\boldsymbol{L}}$ are in the range $[0, 2)$. Therefore, we have $\sigma_{\min}(\boldsymbol{I} + \lambda\hat{\boldsymbol{L}}) = 1$ and $\sigma_{\max}(\boldsymbol{I} + \lambda\hat{\boldsymbol{L}}) \leq 1 + 2\lambda$. In the whole paper we view $\lambda$ as a large value (i.e. $\lambda \gg 1$). In practice, it is possible to use a small $\lambda$, in which case the algorithm works no worse than the case where $\lambda = 1$. Therefore the $\lambda$ used in the paper should actually be understood as $\max\{\lambda, 1\}$. With this assumption, below we assume $\sigma_{\max}(\boldsymbol{I} + \lambda\hat{\boldsymbol{L}}) \leq 3\lambda$ for convenience.

### C.1 DESCENT LEMMA

In this subsection, we introduce a descent lemma that we will use to analyze the convergence rate. This is a standard result in convex optimization, and we refer interested readers to (Bottou et al., 2018) for more details.

**Lemma C.1.** *If $\ell : \mathbb{R}^d \to \mathbb{R}$ is $L$-Lipschitz smooth and $c$-strongly convex, and we have a sequence of points $\left\{\boldsymbol{w}^{(t)}\right\}_{t=1}^{T}$ in $\mathbb{R}^d$ such that*

$$\boldsymbol{w}^{(t+1)} = \boldsymbol{w}^{(t)} - \eta\boldsymbol{g}^{(t)}, \tag{C.1}$$

*where $\left\|\boldsymbol{g}^{(t)} - \nabla\ell\left(\boldsymbol{w}^{(t)}\right)\right\|_2 \leq \gamma\left\|\nabla\ell\left(\boldsymbol{w}^{(t)}\right)\right\|_2$ and $\gamma < 1$, $\eta = \frac{1-\gamma}{(1+\gamma)^2 L}$, then we have*

$$\ell\left(\boldsymbol{w}^{(T)}\right) - \ell^* \leq \left[1 - \kappa^{-1}\left(\frac{1-\gamma}{1+\gamma}\right)^2\right]^T \left[\ell\left(\boldsymbol{w}^{(0)}\right) - \ell^*\right], \tag{C.2}$$

*where $\ell^* = \inf_{\boldsymbol{w}\in\mathbb{R}^d} \ell(\boldsymbol{w})$ and $\kappa = \frac{L}{c}$.*

*Proof.* From the condition $\left\|\boldsymbol{g}^{(t)} - \nabla\ell\left(\boldsymbol{w}^{(t)}\right)\right\|_2 \leq \gamma\left\|\nabla\ell\left(\boldsymbol{w}^{(t)}\right)\right\|_2$, we have that

$$(1 - \gamma)\|\boldsymbol{g}^{(t)}\|_2 \leq \left\|\nabla\ell\left(\boldsymbol{w}^{(t)}\right)\right\|_2 \leq (1 + \gamma)\|\boldsymbol{g}^{(t)}\|_2 \tag{C.3}$$

and

$$-2\left\langle g^{(t)}, \nabla\ell\left(\boldsymbol{w}^{(t)}\right)\right\rangle = \left\|\boldsymbol{g}^{(t)} - \nabla\ell\left(\boldsymbol{w}^{(t)}\right)\right\|_2^2 - \left\|\boldsymbol{g}^{(t)}\right\|_2^2 - \left\|\nabla\ell\left(\boldsymbol{w}^{(t)}\right)\right\|_2^2 \tag{C.4}$$

$$\leq \left[(\gamma^2 - 1) - (1 - \gamma)^2\right]\left\|\nabla\ell\left(\boldsymbol{w}^{(t)}\right)\right\|_2^2 \tag{C.5}$$

$$= 2(\gamma - 1)\left\|\nabla\ell\left(\boldsymbol{w}^{(t)}\right)\right\|_2^2 \tag{C.6}$$

From Lipschitz smoothness, we have

$$\ell\left(\boldsymbol{w}^{(t+1)}\right) - \ell\left(\boldsymbol{w}^{(t)}\right) \leq -\eta\left\langle\boldsymbol{g}^{(t)}, \nabla\ell\left(\boldsymbol{w}^{(t)}\right)\right\rangle + \frac{1}{2}L\eta^2\|\boldsymbol{g}^{(t)}\|_2^2 \tag{C.7}$$

$$\leq \left(\eta(\gamma - 1) + \frac{1}{2}L\eta^2(1 + \gamma)^2\right)\left\|\nabla\ell\left(\boldsymbol{w}^{(t)}\right)\right\|_2^2. \tag{C.8}$$

It's not hard to show that the optimal $\eta$ for eq.(C.8) is $\eta = \frac{1-\gamma}{L(1+\gamma)^2}$. Substituting $\eta = \frac{1-\gamma}{L(1+\gamma)^2}$ to eq.(C.8) and use convexity we can get

$$\ell\left(\boldsymbol{w}^{(t+1)}\right) - \ell^* \leq \left[\ell\left(\boldsymbol{w}^{(t)}\right) - \ell^*\right] - \frac{(1-\gamma)^2}{2L(1+\gamma)^2}\left\|\nabla\ell\left(\boldsymbol{w}^{(t)}\right)\right\|_2^2 \tag{C.9}$$

$$\leq \left[\ell\left(\boldsymbol{w}^{(t)}\right) - \ell^*\right] - \frac{c(1-\gamma)^2}{L(1+\gamma)^2}\left[\ell\left(\boldsymbol{w}^{(t)}\right) - \ell^*\right] \tag{C.10}$$

By induction we have

$$\ell\left(\boldsymbol{w}^{(T)}\right) - \ell^* \leq \left[1 - \kappa^{-1}\left(\frac{1-\gamma}{1+\gamma}\right)^2\right]^T \left[\ell\left(\boldsymbol{w}^{(0)}\right) - \ell^*\right]. \tag{C.11}$$

$\square$

## C.2 BOUND OF LOSS VALUE BY GRADIENT

In this subsection, we derive an inequality that allows us to bound the value of loss function by the norm of gradient.

**Lemma C.2.** *If $\ell : \mathbb{R}^d \to \mathbb{R}$ is a $\mathscr{C}$-strongly convex and smooth function, and $\boldsymbol{w}^* = \arg\min_{\boldsymbol{w}\in\mathbb{R}^d}\ell(\boldsymbol{w})$ is a global optimal, then for any $\boldsymbol{w} \in \mathbb{R}^d$ and $\epsilon \in (0,1)$, we have*

$$\ell(\boldsymbol{w}) \leq \max\left\{(1+\epsilon)\ell(\boldsymbol{w}^*), \frac{1}{\epsilon\mathscr{C}}\|\nabla\ell(\boldsymbol{w})\|^2\right\}. \tag{C.12}$$

*Proof.* Using the inequality (4.12) from (Bottou et al., 2018), we have

$$\ell(\boldsymbol{w}) \leq \frac{1}{2\mathscr{C}}\|\nabla\ell(\boldsymbol{w})\|^2 + \ell(\boldsymbol{w}^*). \tag{C.13}$$

If $\ell(\boldsymbol{w}) \geq (1+\epsilon)\ell(\boldsymbol{w}^*)$, we have

$$\ell(\boldsymbol{w}) \leq \frac{1}{2\mathscr{C}}\|\nabla\ell(\boldsymbol{w})\|^2 + \frac{1}{1+\epsilon}\ell(\boldsymbol{w}). \tag{C.14}$$

Shifting the terms in eq.(C.14) gives $\ell(\boldsymbol{w}) \leq \frac{1+\epsilon}{2\mathscr{C}\epsilon}\|\nabla\ell(\boldsymbol{w})\|^2 \leq \frac{1}{\epsilon\mathscr{C}}\|\nabla\ell(\boldsymbol{w})\|^2$, which proves the claim.

$\square$

## C.3 PROOF OF LEMMA 5.1

Given a point $\boldsymbol{w} \in \mathbb{R}^d$, we first consider the error between the estimated gradient $\widetilde{\nabla\ell_i(\boldsymbol{w})}$ and the true gradient $\nabla\ell_i(\boldsymbol{w})$. Below we denote $\boldsymbol{H} = \boldsymbol{I} + \lambda\hat{\boldsymbol{L}}$ and $\boldsymbol{z} = \boldsymbol{X}\boldsymbol{w} - \hat{\boldsymbol{y}}_i$. It's not hard to notice that $\ell_i(\boldsymbol{w}) = \frac{1}{2}\left\|\boldsymbol{H}^{-1}\boldsymbol{z}\right\|^2$ and $\nabla\ell_i(\boldsymbol{w}) = \boldsymbol{X}^\top\boldsymbol{H}^{-2}\boldsymbol{z}$. Moreover, we denote $\ell_i^* = \inf_{\boldsymbol{w}}\ell_i^*$ as the optimal value of $\ell_i$.

we have

$$\left\|\widetilde{\nabla\ell_i(\boldsymbol{w})} - \nabla\ell_i(\boldsymbol{w})\right\| = \left\|\boldsymbol{X}^\top\left[\mathcal{S}\left(\boldsymbol{H}^{-1}\boldsymbol{z}\right) - \boldsymbol{H}^{-2}\boldsymbol{z} + \mathcal{S}(\mathcal{S}(\boldsymbol{z})) - \mathcal{S}\left(\boldsymbol{H}^{-1}\boldsymbol{z}\right)\right]\right\| \tag{C.15}$$

$$\leq \left\|\boldsymbol{X}^\top\left[\mathcal{S}\left(\boldsymbol{H}^{-1}\boldsymbol{z}\right) - \boldsymbol{H}^{-2}\boldsymbol{z}\right]\right\| + \left\|\boldsymbol{X}^\top\left[\mathcal{S}(\mathcal{S}(\boldsymbol{z}) - \boldsymbol{H}^{-1}\boldsymbol{z})\right]\right\|. \tag{C.16}$$

As shown, the gradient error can be decomposed into two terms, namely $\left\|\widetilde{\nabla\ell_i(\boldsymbol{w})} - \nabla\ell_i(\boldsymbol{w})\right\| = E_1 + E_2$, where $E_1 = \left\|\boldsymbol{X}^\top\left[\mathcal{S}\left(\boldsymbol{H}^{-1}\boldsymbol{z}\right) - \boldsymbol{H}^{-2}\boldsymbol{z}\right]\right\|$ and $E_2 = \left\|\boldsymbol{X}^\top\left[\mathcal{S}(\mathcal{S}(\boldsymbol{z}) - \boldsymbol{H}^{-1}\boldsymbol{z})\right]\right\|$. Below we analyze each part separately.

Notice that, as we assumed $\boldsymbol{X}$ is full-rank, $\ell_i$ is a strongly convex function with parameter $\mathscr{C} = \sigma_{\min}\left(\boldsymbol{X}^\top\boldsymbol{H}^{-2}\boldsymbol{X}\right) \geq \frac{\sigma_{\min}(\boldsymbol{X})^2}{9\lambda^2}$. Let $\mathcal{D} = \left\{\boldsymbol{w} \in \mathbb{R}^d \middle| \ell_i(\boldsymbol{w}) \leq (1+\epsilon)\ell_i^*\right\}$. If $\boldsymbol{w} \in \mathcal{D}$, then $\boldsymbol{w}$ is

already a good enough solution. Below we assume $\boldsymbol{w} \notin \mathcal{D}$, in which case by Lemma C.2, we have $\ell_i(\boldsymbol{w}) \leq \frac{1}{\epsilon \mathscr{C}} \|\nabla \ell_i(\boldsymbol{w})\|^2$, which means

$$\left\|\boldsymbol{H}^{-1}\boldsymbol{z}\right\|^2 \leq \frac{2}{\mathscr{C}\epsilon}\left\|\boldsymbol{X}^{-1}\boldsymbol{H}^{-2}\boldsymbol{z}\right\|^2 \leq \frac{18\lambda^2}{\sigma_{\min}(\boldsymbol{X})^2\epsilon}. \tag{C.17}$$

For $E_1$, we have

$$E_1 = \left\|\boldsymbol{X}^\top \left[\mathcal{S}\left(\boldsymbol{H}^{-1}\boldsymbol{z}\right) - \boldsymbol{H}^{-2}\boldsymbol{z}\right]\right\|_2 \tag{C.18}$$

$$\leq \sigma_{\max}(\boldsymbol{X})\left\|\mathcal{S}\left(\boldsymbol{H}^{-1}\boldsymbol{z}\right) - \boldsymbol{H}^{-2}\boldsymbol{z}\right\|_2 \tag{C.19}$$

$$\leq \sigma_{\max}(\boldsymbol{X})\left\|\mathcal{S}\left(\boldsymbol{H}^{-1}\boldsymbol{z}\right) - \boldsymbol{H}^{-2}\boldsymbol{z}\right\|_H \tag{C.20}$$

$$\leq \sigma_{\max}(\boldsymbol{X})\mu\left\|\boldsymbol{H}^{-2}\boldsymbol{z}\right\|_H \tag{C.21}$$

$$\leq \sigma_{\max}(\boldsymbol{X})\mu\left\|\boldsymbol{H}^{-1}\boldsymbol{z}\right\|_2 \tag{C.22}$$

$$\leq \sqrt{\frac{18}{\epsilon}}\kappa(\boldsymbol{X})\lambda\mu\left\|\boldsymbol{X}^\top\boldsymbol{H}^{-2}\boldsymbol{z}\right\|_2 \tag{C.23}$$

$$\leq 5\epsilon^{-1/2}\kappa(\boldsymbol{X})\lambda\mu\|\nabla\ell_i(\boldsymbol{w})\|_2. \tag{C.24}$$

For $E_2$, we have

$$E_2 = \left\|\boldsymbol{X}^\top\left[\mathcal{S}(\mathcal{S}(\boldsymbol{z}) - \boldsymbol{H}^{-1}\boldsymbol{z})\right]\right\|_2 \tag{C.25}$$

$$\leq \sigma_{\max}(\boldsymbol{X})(1+\mu)\|\boldsymbol{H}^{-1}\left(\mathcal{S}(\boldsymbol{z}) - \boldsymbol{H}^{-1}\boldsymbol{z}\right)\|_H \tag{C.26}$$

$$\leq \sigma_{\max}(\boldsymbol{X})(1+\mu)\|\mathcal{S}(\boldsymbol{z}) - \boldsymbol{H}^{-1}\boldsymbol{z}\|_H \tag{C.27}$$

$$\leq \sigma_{\max}(\boldsymbol{X})(1+\mu)\mu\|\boldsymbol{H}^{-1}\boldsymbol{z}\|_H \tag{C.28}$$

$$\leq \sqrt{\frac{18}{\epsilon}}\kappa(\boldsymbol{X})(1+\mu)\mu\lambda\sqrt{3\lambda}\|\boldsymbol{X}^\top\boldsymbol{H}^{-2}\boldsymbol{z}\|_2 \tag{C.29}$$

$$\leq 20\epsilon^{-1/2}\kappa(\boldsymbol{X})\mu\lambda^2\|\nabla\ell_i(\boldsymbol{w}_i)\|_2. \tag{C.30}$$

Combine the bound of $E_1$ and $E_2$, we have

$$\left\|\widetilde{\nabla\ell_i(\boldsymbol{w}_i)} - \nabla\ell_i(\boldsymbol{w}_i)\right\| \leq E_1 + E_2 \leq 25\epsilon^{-1/2}\kappa(\boldsymbol{X})\mu\lambda^2\|\nabla\ell_i(\boldsymbol{w}_i)\|_2. \tag{C.31}$$

Now, consider the optimization. Let $\left\{\boldsymbol{w}_i^{(t)}\right\}_{t=1}^T$ be a sequence such that

$$\boldsymbol{w}_i^{(t+1)} = \boldsymbol{w}^{(t)} - \eta\widetilde{\nabla\ell_i\left(\boldsymbol{w}_i^{(t)}\right)}. \tag{C.32}$$

Let $\kappa$ be the condition number of $\boldsymbol{X}^\top\boldsymbol{H}^{-2}\boldsymbol{X}$ and let $\ell_i^*$ be the optimal value of $\ell_i$. Let $\gamma = 25\epsilon^{-1/2}\kappa(\boldsymbol{X})\lambda^2\mu \leq \frac{1}{2}$. From Lemma C.1, we have with a proper value of $\eta$,

$$\ell_i\left(\boldsymbol{w}_i^{(T)}\right) - \ell^* \leq \left(1 - \left(\frac{1-\gamma}{1+\gamma}\right)^2\kappa^{-1}\right)^T\left[\ell_i\left(\boldsymbol{w}_i^{(0)}\right) - \ell_i^*\right] \tag{C.33}$$

$$\leq \left(1 - (9\kappa)^{-1}\right)^T\left[\ell_i\left(\boldsymbol{w}_i^{(0)}\right) - \ell_i^*\right]. \tag{C.34}$$

Therefore, the number of iterations $T$ required to achieve $\epsilon$ error rate is

$$T = O\left(\left(\log\frac{1}{1 - (9\kappa)^{-1}}\right)^{-1}\log\frac{1}{\epsilon}\right) = O\left(\kappa\log\frac{1}{\epsilon}\right). \tag{C.35}$$

## C.4 PROOF OF LEMMA 5.2

The basic idea of Algorithm 2 is to use ridge leverage score sampling methods to obtain a spectral sparsifier. Let $\hat{\boldsymbol{B}} = \boldsymbol{B}\boldsymbol{D}^{-1/2}$ be the normalized incidence matrix and $\hat{\boldsymbol{b}}_i$ the $i$-th row of $\hat{\boldsymbol{B}}$. Given $\lambda^{-1} > 0$, for $i \in \{1, 2, \cdots, m\}$, the $i$-th ridge leverage score is defined as

$$l_i := \hat{\boldsymbol{b}}_i^\top (\hat{\boldsymbol{L}} + \lambda^{-1}\boldsymbol{I})^{-1}\hat{\boldsymbol{b}}_i \tag{C.36}$$

$$= \hat{\boldsymbol{b}}_i^\top (\hat{\boldsymbol{L}} + \lambda^{-1}\boldsymbol{I})^{-1}\hat{\boldsymbol{L}}(\hat{\boldsymbol{L}} + \lambda^{-1}\boldsymbol{I})^{-1}\hat{\boldsymbol{b}}_i + \lambda^{-1}\hat{\boldsymbol{b}}_i^\top (\hat{\boldsymbol{L}} + \lambda^{-1}\boldsymbol{I})^{-2}\hat{\boldsymbol{b}}_i \tag{C.37}$$

$$= \|\hat{\boldsymbol{B}}(\hat{\boldsymbol{L}} + \lambda^{-1}\boldsymbol{I})^{-1}\hat{\boldsymbol{b}}_i\|^2 + \lambda^{-1}\|(\hat{\boldsymbol{L}} + \lambda^{-1}\boldsymbol{I})^{-1}\hat{\boldsymbol{b}}_i\|^2. \tag{C.38}$$

It is not affordable to compute all the $m$ ridge leverage scores exactly. Therefore, we first use Johnson–Lindenstrauss lemma to reduce the dimension. Recall that in Algorithm 2 we define $\boldsymbol{\Pi}_1 \in \mathbb{R}^{k \times m}$ and $\boldsymbol{\Pi}_2 \in \mathbb{R}^{k \times n}$ to be Gaussian sketches (that is, each entry of the matrices are i.i.d Gaussian random variables $\mathcal{N}(0, 1/k)$). We set $k = O(\log m)$, and by using Lemma C.3 we have the following claim holds with probability at least $1 - \frac{1}{8n}$:

$$\|\boldsymbol{\Pi}_1\hat{\boldsymbol{B}}(\hat{\boldsymbol{L}} + \lambda^{-1}\boldsymbol{I})^{-1}\hat{\boldsymbol{b}}_i\| \approx_{2^{1/4}-1} \|\hat{\boldsymbol{B}}(\hat{\boldsymbol{L}} + \lambda^{-1}\boldsymbol{I})^{-1}\hat{\boldsymbol{b}}_i\| \quad \text{for all } i \in \{1, 2, \cdots, m\}. \tag{C.39}$$

Similarly, the following claim also holds with probability at least $1 - \frac{1}{8n}$:

$$\|\boldsymbol{\Pi}_2(\hat{\boldsymbol{L}} + \lambda^{-1}\boldsymbol{I})^{-1}\hat{\boldsymbol{b}}_i\| \approx_{2^{1/4}-1} \|(\hat{\boldsymbol{L}} + \lambda^{-1}\boldsymbol{I})^{-1}\hat{\boldsymbol{b}}_i\| \quad \text{for all } i \in \{1, 2, \cdots, m\}. \tag{C.40}$$

By summing over above two inequalities (after squared) and taking an union bound, with probability $1 - \frac{1}{4n}$ we have

$$\|\boldsymbol{\Pi}_1\hat{\boldsymbol{B}}(\hat{\boldsymbol{L}} + \lambda^{-1}\boldsymbol{I})^{-1}\hat{\boldsymbol{b}}_i\|^2 + \lambda^{-1}\|\boldsymbol{\Pi}_2(\hat{\boldsymbol{L}} + \lambda^{-1}\boldsymbol{I})^{-1}\hat{\boldsymbol{b}}_i\|^2 \approx_{\sqrt{2}-1} l_i \quad \text{for all } \ i \in \{1, 2, \cdots, m\}. \tag{C.41}$$

However it is still too expensive to compute $\boldsymbol{\Pi}_1\hat{\boldsymbol{B}}(\hat{\boldsymbol{L}} + \lambda^{-1}\boldsymbol{I})^{-1}$ and $\boldsymbol{\Pi}_2(\hat{\boldsymbol{L}} + \lambda^{-1}\boldsymbol{I})^{-1}$, since computing $(\hat{\boldsymbol{L}} + \lambda^{-1}\boldsymbol{I})^{-1}$ itself takes prohibitive $O(n^3)$ time. Instead, notice that $\hat{\boldsymbol{L}} + \lambda^{-1}\boldsymbol{I}$ is a SDD matrix, thus we can apply the SDD solver to the $k$ columns of matrix $(\boldsymbol{\Pi}_1\hat{\boldsymbol{B}})^\top$ and $(\boldsymbol{\Pi}_2)^\top$ respectively. According to Lemma A.1, there is a linear operator $\mathsf{SSolve}_\epsilon(\hat{\boldsymbol{L}} + \lambda^{-1}\boldsymbol{I}; \boldsymbol{x})$ that runs in time $\tilde{O}(\mathrm{nnz}(\hat{\boldsymbol{L}} + \lambda^{-1}\boldsymbol{I}) \cdot \log 1/\epsilon) = \tilde{O}(m \log 1/\epsilon)$ such that for any $\boldsymbol{x}^\top \in \mathbb{R}^n$, it outputs $\tilde{\boldsymbol{x}}$ that satisfies $\|\tilde{\boldsymbol{x}} - \boldsymbol{x}^\top(\hat{\boldsymbol{L}} + \lambda^{-1}\boldsymbol{I})^{-1}\|_{\hat{\boldsymbol{L}}+\lambda^{-1}\boldsymbol{I}} \leq \epsilon\|\boldsymbol{x}^\top(\hat{\boldsymbol{L}} + \lambda^{-1})\|_{\hat{\boldsymbol{L}}+\lambda^{-1}\boldsymbol{I}}$. For our purpose we set $\epsilon = 2^{1/4}$. Denote $\boldsymbol{B}_{\mathcal{S}}$ as the matrix obtained by applying each column of $(\boldsymbol{\Pi}_1\hat{\boldsymbol{B}})^\top$ to $\mathsf{SSolve}_\epsilon(\hat{\boldsymbol{L}} + \lambda^{-1}\boldsymbol{I}; \boldsymbol{x})$, that is, the $j$-th row of $\mathsf{SSolve}_{2^{1/4}}(\hat{\boldsymbol{L}} + \lambda^{-1}\boldsymbol{I}; (\boldsymbol{\Pi}_1\hat{\boldsymbol{B}})_j^\top)$. Similarly we denote $\boldsymbol{\Pi}_{\mathcal{S}}$ as the matrix with $j$-th row equals to $\mathsf{SSolve}_{2^{1/4}}(\hat{\boldsymbol{L}} + \lambda^{-1}\boldsymbol{I}; (\boldsymbol{\Pi}_2)_j^\top)$. By using Lemma A.1 to both solvers we have

$$\|\boldsymbol{B}_{\mathcal{S}}\hat{\boldsymbol{b}}_i\|^2 + \lambda^{-1}\|\boldsymbol{\Pi}_{\mathcal{S}}\hat{\boldsymbol{b}}_i\|^2 \approx_{\sqrt{2}-1} \|\boldsymbol{\Pi}_1\hat{\boldsymbol{B}}(\hat{\boldsymbol{L}} + \lambda^{-1}\boldsymbol{I})^{-1}\hat{\boldsymbol{b}}_i\|^2 + \lambda^{-1}\|\boldsymbol{\Pi}_2(\hat{\boldsymbol{L}} + \lambda^{-1}\boldsymbol{I})^{-1}\hat{\boldsymbol{b}}_i\|^2 \tag{C.42}$$

for all $i \in \{1, 2, \cdots, m\}$. By eq.(C.41) and eq.(C.42), if we set $\tilde{l}_i := \|\boldsymbol{B}_{\mathcal{S}}\hat{\boldsymbol{b}}_i\|^2 + \lambda^{-1}\|\boldsymbol{\Pi}_{\mathcal{S}}\hat{\boldsymbol{b}}_i\|^2$, then with probability $1 - \frac{1}{4n}$, we obtain all the approximation of ridge leverage scores $\{\tilde{l}_i\}_{i=1}^m$ such that $\tilde{l}_i \approx_{1/2} l_i$ holds for all $i$. Notice that since $\boldsymbol{B}_{\mathcal{S}}, \boldsymbol{\Pi}_{\mathcal{S}} \in \mathbb{R}^{k \times n}$, and that each $\hat{\boldsymbol{b}}_i$ only contains 2 non-zero entries, thus it takes $\tilde{O}(k \cdot m)$ to pre-compute $\boldsymbol{B}_{\mathcal{S}}$ and $\boldsymbol{\Pi}_{\mathcal{S}}$, and takes $O(2k \cdot m)$ to compute $\{\boldsymbol{B}_{\mathcal{S}}\hat{\boldsymbol{b}}_i, \boldsymbol{\Pi}_{\mathcal{S}}\hat{\boldsymbol{b}}_i\}_{i=1}^m$. To summarize, computing all $\tilde{l}_i$ takes $\tilde{O}(km) = \tilde{O}(m \log m) = \tilde{O}(m)$. With these ridge leverage score approximations, we apply Lemma C.4 to matrix $\hat{\boldsymbol{B}}$ with choice $\delta = 1/4n$. By setting $\tilde{\boldsymbol{L}} := \hat{\boldsymbol{B}}^\top \boldsymbol{S}^\top \boldsymbol{S}\hat{\boldsymbol{B}}$ we have $\tilde{\boldsymbol{L}} + \lambda^{-1}\boldsymbol{I} \approx_\epsilon \hat{\boldsymbol{L}} + \lambda^{-1}\boldsymbol{I}$ holds with probability $1 - 1/4n$. By applying another union bound we obtain our final result:

$$\tilde{\boldsymbol{L}} + \lambda^{-1}\boldsymbol{I} \approx_\epsilon \hat{\boldsymbol{L}} + \lambda^{-1}\boldsymbol{I} \quad \text{with probability} \ \ 1 - \frac{1}{2n}. \tag{C.43}$$

Finally, according to Lemma C.4, the number of edges of $\tilde{\boldsymbol{L}}$ is $s = Cn_\lambda \log(n^2)/\epsilon^2 = O(n_\lambda \log n/\epsilon^2)$. Since the last step of computing $\tilde{\boldsymbol{L}} = (\boldsymbol{S}\hat{\boldsymbol{B}})^\top(\boldsymbol{S}\hat{\boldsymbol{B}})$ only takes $O(s) = \tilde{O}(n_\lambda/\epsilon^2)$ due to the sparsity of $\hat{\boldsymbol{B}}$, the overall time complexity of Algorithm 2 is $\tilde{O}(m + n_\lambda/\epsilon^2)$.

**Lemma C.3** (Johnson–Lindenstrauss, Dasgupta & Gupta (2003)). *Let $\mathbf{\Pi} \in \mathbb{R}^{k \times n}$ be Gaussian sketch matrix with each entry independent and equal to $\mathcal{N}(0, 1/k)$ where $\mathcal{N}(0, 1)$ denotes a standard Gaussian random variable. If we choose $k = O\left(\frac{\log(1/\delta)}{\epsilon^2}\right)$, then for any vector $\boldsymbol{x} \in \mathbb{R}^n$, with probability $1 - \delta$ we have*

$$(1 - \epsilon)\|\boldsymbol{x}\| \leq \|\mathbf{\Pi}\boldsymbol{x}\| \leq (1 + \epsilon)\|\boldsymbol{x}\|. \tag{C.44}$$

**Lemma C.4** (Spectral approximation, Cohen et al. (2017b)). *Let $\boldsymbol{S}$ be an $s \times m$ subsampling matrix with probabilities $p_i = \tilde{\ell}_i/Z$ where $\tilde{\ell}_i \approx_{1/2} \ell_i$ and $Z$ is the normalization constant. If we have $s \geq Cn_\lambda \log(n/\delta)/\epsilon^2$ for some constant $C > 0$ and $\epsilon, \delta \in (0, 1/2]$, then we have*

$$\hat{\boldsymbol{B}}^\top \boldsymbol{S}^\top \boldsymbol{S} \hat{\boldsymbol{B}} + \lambda^{-1}\boldsymbol{I} \approx_\epsilon \hat{\boldsymbol{B}}^\top \hat{\boldsymbol{B}} + \lambda^{-1}\boldsymbol{I} \tag{C.45}$$

*holds with probability $1 - \delta$. Here $n_\lambda = \text{Tr}[\hat{\boldsymbol{L}}(\hat{\boldsymbol{L}} + \lambda^{-1}\boldsymbol{I})^{-1}]$.*

## C.5 PROOF OF LEMMA 5.3

We first prove a lemma which is related to the approximation rate of squared matrices.

**Lemma C.5.** *Suppose that $\mathbf{\Sigma}$ and $\widetilde{\mathbf{\Sigma}}$ are two $n \times n$ PD matrices, and $\widetilde{\mathbf{\Sigma}} \approx_{\frac{\beta}{\kappa}} \mathbf{\Sigma}$, where $\kappa$ is the condition number of $\mathbf{\Sigma}$ and $\beta \in \left(0, \frac{1}{8}\right)$, then we have*

$$\widetilde{\mathbf{\Sigma}}^2 \approx_{8\beta} \mathbf{\Sigma}^2. \tag{C.46}$$

*Proof.* Let $\epsilon = \frac{\beta}{\kappa}$. The condition $\mathbf{\Sigma} \approx_\epsilon \widetilde{\mathbf{\Sigma}}$ implies that $\left\|\mathbf{\Sigma} - \widetilde{\mathbf{\Sigma}}\right\| \leq \epsilon \|\mathbf{\Sigma}\|$. We have

$$\left\|\left(\mathbf{\Sigma} - \widetilde{\mathbf{\Sigma}}\right)\boldsymbol{x}\right\| = \left\|\left(\mathbf{\Sigma} - \widetilde{\mathbf{\Sigma}}\right)\mathbf{\Sigma}^{-1}\mathbf{\Sigma}\boldsymbol{x}\right\| \tag{C.47}$$

$$\leq \left\|\left(\mathbf{\Sigma} - \widetilde{\mathbf{\Sigma}}\right)\mathbf{\Sigma}^{-1}\right\| \times \|\mathbf{\Sigma}\boldsymbol{x}\| \tag{C.48}$$

$$\leq \left\|\left(\mathbf{\Sigma} - \widetilde{\mathbf{\Sigma}}\right)\right\| \left\|\mathbf{\Sigma}^{-1}\right\| \times \|\mathbf{\Sigma}\boldsymbol{x}\| \tag{C.49}$$

$$\leq \epsilon \|\mathbf{\Sigma}\| \left\|\mathbf{\Sigma}^{-1}\right\| \times \|\mathbf{\Sigma}\boldsymbol{x}\| \tag{C.50}$$

$$= \epsilon\kappa \|\mathbf{\Sigma}\boldsymbol{x}\| \tag{C.51}$$

$$= \beta \|\mathbf{\Sigma}\boldsymbol{x}\|. \tag{C.52}$$

For any $\boldsymbol{x} \in \mathbb{R}^n$, by triangle inequality we have

$$\|\mathbf{\Sigma}\boldsymbol{x}\| - \left\|\left(\mathbf{\Sigma} - \widetilde{\mathbf{\Sigma}}\right)\boldsymbol{x}\right\| \leq \left\|\widetilde{\mathbf{\Sigma}}\boldsymbol{x}\right\| \leq \|\mathbf{\Sigma}\boldsymbol{x}\| + \left\|\left(\mathbf{\Sigma} - \widetilde{\mathbf{\Sigma}}\right)\boldsymbol{x}\right\|. \tag{C.53}$$

Subtracting the inequality $\left\|\left(\mathbf{\Sigma} - \widetilde{\mathbf{\Sigma}}\right)\boldsymbol{x}\right\| \leq \beta \|\mathbf{\Sigma}\boldsymbol{x}\|$ we derived before into eq.(C.53) and squaring all sides, we have

$$(1 - \beta)^2 \boldsymbol{x}^\top \mathbf{\Sigma}^2 \boldsymbol{x} \leq \boldsymbol{x}^\top \widetilde{\mathbf{\Sigma}} \boldsymbol{x} \leq (1 + \beta)^2 \boldsymbol{x}^\top \mathbf{\Sigma}^2 \boldsymbol{x}. \tag{C.54}$$

Since $0 < \beta < \frac{1}{8}$, the claim is proved. $\qquad\square$

Let $\boldsymbol{T} = \boldsymbol{X}^\top \left(\boldsymbol{I} + \lambda\hat{\boldsymbol{L}}\right)^{-2} \boldsymbol{X}$ be the true Hessian. Let $\tilde{\boldsymbol{T}} = \boldsymbol{X}^\top \left(\boldsymbol{I} + \lambda\tilde{\boldsymbol{L}}\right)^{-2} \boldsymbol{X}$ be the approximated Hessian using sparsified Laplacian $\tilde{\boldsymbol{L}}$. By Lemma 5.2, $\boldsymbol{I} + \lambda\tilde{\boldsymbol{L}} \approx_{\frac{\beta}{3\lambda}} \boldsymbol{I} + \lambda\boldsymbol{L}$ with probability at least $1 - \frac{1}{2n}$. From Lemma C.5, we have $\tilde{\boldsymbol{T}} \approx_{8\beta} \boldsymbol{T}$.

Next, we show that $\boldsymbol{Q}^\top \boldsymbol{Q} \approx_{O(\beta)} \boldsymbol{T}$. Let $\tilde{\boldsymbol{H}} = \boldsymbol{I} + \lambda\tilde{\boldsymbol{L}}$. Let $\mathcal{S}$ be the operator defined by $\text{SSolve}_{\frac{\beta}{\sqrt{3\lambda}}}\left(\tilde{\boldsymbol{H}}, \cdot\right)$, i.e. $\boldsymbol{q}_j = \mathcal{S}(\boldsymbol{x}_j)$ where $\boldsymbol{q}_j$ and $\boldsymbol{x}_j$ are the $j$-th column of $\boldsymbol{Q}$ and $\boldsymbol{X}$ respectively.

From Lemma A.1 we have for any $z \in \mathbb{R}^d$,

$$\left\| Qz - \tilde{H}^{-1} X z \right\|_2^2 = \left\| \sum_{j=1}^d z_j \left( \mathcal{S}(x_j) - \tilde{H} x_j \right) \right\|_2^2 \tag{C.55}$$

$$= \left\| \mathcal{S} \left( \sum_{j=1}^d z_j x_j \right) - \tilde{H} \left( \sum_{j=1}^d z_j x_j \right) \right\|_2^2 \tag{C.56}$$

$$\leq \frac{\beta^2}{3\lambda} \left\| \tilde{H}^{-1} X z \right\|_{\tilde{H}}^2 \tag{C.57}$$

$$\leq \beta^2 \| \tilde{H}^{-1} X z \|_2^2. \tag{C.58}$$

Therefore we have

$$(1-\beta)\| \tilde{H}^{-1} X z \|_2 \leq \| Q z \| \leq (1+\beta)\| \tilde{H}^{-1} X z \|_2, \tag{C.59}$$

and this is equivalent to

$$(1-\beta)^2 z^\top (X \tilde{H}^{-2} X) z \leq z^\top Q^\top Q z \leq (1+\beta)^2 z^\top \left( X^\top \tilde{H}^{-2} X \right) z. \tag{C.60}$$

Notice that $X^\top \tilde{H}^{-2} X = \tilde{T}$. We conclude that $Q^\top Q \approx_{2\beta + \beta^2} \tilde{T}$. When $\beta < \frac{1}{8}$, we have $Q^\top Q \approx_{4\beta} \tilde{T}$.

Lastly, from Lemma A.2 and the discussions in Appendix A.3, we have there exists a constant $C' \in (0, 1/4)$, such that

$$P = \tilde{Q}^\top \tilde{Q} \approx_{C'} Q^\top Q \tag{C.61}$$

with probability at least $1 - \frac{1}{2n}$.

Put the results above together and we get

$$P \approx_{12\beta + C'} T. \tag{C.62}$$

Notice that $12\beta + C' < \frac{1}{2}$. Using a union bound can prove that the fail probability of this whole process is bounded by $\frac{1}{n}$.

## C.6 Proof of Lemma 5.4

Let $F = (I + \lambda \hat{L}) X$. We have the Hessian of $\ell'$ is

$$\nabla^2 \ell' = P^{-\frac{1}{2}} F^\top F P^{-\frac{1}{2}}. \tag{C.63}$$

From the condition that $P \approx_{c_0} F^\top F$, we have $F^\top F \preceq (1 + c_0) P$, which implies

$$P^{-1/2} F^\top F P^{-1/2} \preceq (1 + c_0) I. \tag{C.64}$$

Similarly, since $P \preceq (1 + c_0) F^\top F$, we have

$$\left( F^\top F \right)^{-1} \preceq (1 + c_0) P^{-1}, \tag{C.65}$$

which implies

$$\left[ \lambda_{\min}(P^{-1/2} F^\top F P^{-1/2}) \right]^{-1} = \lambda_{\max} \left( P^{1/2} \left( F^\top F \right)^{-1} P^{1/2} \right) \tag{C.66}$$

$$\leq 1 + c_0. \tag{C.67}$$

.

Notice that $\ell'$ and $\ell$ have the same global optimal value, let it be $\ell^*$. For any $w'$ satisfies $\ell'(w') - \ell^* \leq \gamma$, we have

$$\ell \left( P^{1/2} w' \right) - \ell^* = \ell'(w') - \ell^* \leq \gamma. \tag{C.68}$$

Therefore, if $w'$ is a solution of $\ell'$ with $\epsilon$ error rate, then $P^{1/2} w'$ is a solution of $\ell$ with $\epsilon$ error rate.

### C.7    PROOF OF THEOREM 5.1

As we noted in the main paper, Algorithm 1 is composed by two components: constructing the preconditioner $\boldsymbol{P}$ and applying it to the optimization.

**Constructing the Preconditioner.**    From Lemma 5.2, the first step that applies spectral sparsifier to get $\tilde{\boldsymbol{L}}$ requires $\tilde{O}(m)$ time and the number of non-zero entries in $\tilde{\boldsymbol{L}}$ is $\tilde{O}\left(n_\lambda \lambda^2\right)$. By Lemma A.1, running the SDD solver with error rate $\frac{\beta}{3\lambda}$ in the second step requires $\tilde{O}\left(n_\lambda \lambda^2\right)$ time. Notice that since $\boldsymbol{X}$ is an $n \times d$ matrix, we actually need to run SDD solver for $d$ times, and this introduces another $d$ factor in the time complexity.

As we noted in Appendix A.3, applying the Hadamard transformation to $\boldsymbol{RQ}$ requires $\tilde{O}(d)$ time, and applying the subsampling requires $\tilde{O}(n)$ time. Since $\tilde{\boldsymbol{Q}} \in \mathbb{R}^{s \times d}$, calculating $\boldsymbol{P} = \tilde{\boldsymbol{Q}}^\top \boldsymbol{Q}$ requires $O(d^2 s)$ time. Since $s = \tilde{O}(d)$, this step takes $\tilde{O}(d^3)$ time. We use brute force to calculate $\boldsymbol{P}' = \boldsymbol{P}^{-1/2}$, and this takes $O(d^3)$ time.

As a summary, constructing the preconditioner $\boldsymbol{P}'$ takes $\tilde{O}(n_\lambda \lambda^2 d + nd + d^3)$ time. Notice that we only perform this step once during the whole algorithm.

**Performing the Iterations.**    Next we consider the time required for each iteration. We calculate $\boldsymbol{u}^{(t)}$ from right to left and it takes $O(nd)$ time. Next, we need to perform two SDD solvers with error rate $\mu$. From Lemma A.1, it takes $\tilde{O}\left[m \log\left(\frac{1}{\mu}\right)\right]$, which is $\tilde{O}\left[m \log\left(\frac{1}{\epsilon}\right)\right]$. Calculating $\boldsymbol{g}^{(t)}$ and $\boldsymbol{w}^{(t)}$ is straight-forward and takes $\tilde{O}(nd)$ time.

To conclude, performing each iteration requires $\tilde{O}\left[m \log\left(\frac{1}{\epsilon}\right) + nd\right]$ time. By Lemma 5.1 and Lemma 5.3, with a proper step size, the number of iterations needed for solving outer problem is $\tilde{O}\left(\log 1/\epsilon\right)$.

Combining the analysis above together, to overall complexity is

$$\tilde{O}\left[n_\lambda \lambda^2 d + nd + d^3 + \left(m \log\left(\frac{1}{\epsilon}\right) + nd\right) \log\left(\frac{1}{\epsilon}\right)\right] = \tilde{O}\left(n_\lambda \lambda^2 d + d^3 + (m + nd)(\log 1/\epsilon)^2\right), \tag{C.69}$$

which proves the claim.

## D    FURTHER DISCUSSIONS

In this section, we extend some of the discussions in the main paper.

### D.1    AN ANALYSIS OF CE LOSS V.S. MSE LOSS

Although HERTA is derived from MSE loss, the experiment result shows it also works on CE loss. In this subsection we provide an analysis showing the similarity of the gradient and Hessian of TWIRLS on CE loss and MSE loss, to offer a intuitive explanation why a method that is derived from MSE loss can work on CE loss.

In this section, for a node $u \in \{1, 2, \cdots, n\}$, we use $\boldsymbol{y}^{(u)} \in \mathbb{R}^c$ to represent the $u$-th row of $\boldsymbol{Y}$ (notice we use super-script here to distinguish from $\boldsymbol{y}_i$ used before), $\boldsymbol{h}_u \in \mathbb{R}^n$ to represent the $u$-th row of $\left(\boldsymbol{I} + \lambda \hat{\boldsymbol{L}}\right)^{-1}$. For $p \in \{1, 2, \cdots d\}$, we use $\boldsymbol{x}_p \in \mathbb{R}^n$ to denote the $p$-th column of $\boldsymbol{X}$. For $i \in \{1, 2, \cdots c\}$, we use $y_i^{(u)}$ to denote the $i$-th entry of $\boldsymbol{y}^{(u)}$ (or in other words the $(u, i)$-th entry of $\boldsymbol{Y}$), and $w_{p,i}$ to denote the $(p, i)$-th entry of $\boldsymbol{W}$.

The MSE loss of TWIRLS can be decomposed into summation of sub-losses of each node, i.e.

$$\ell(\boldsymbol{W}) = \sum_{u=1}^{n} \frac{1}{2}\left\|\boldsymbol{h}_u^\top \boldsymbol{X} \boldsymbol{W} - \boldsymbol{y}^{(u)}\right\|_{\mathcal{F}}^2 = \sum_{u=1}^{n} \ell^{(u)}(\boldsymbol{W}), \tag{D.1}$$

where $\ell^{(u)}(\boldsymbol{W}) = \frac{1}{2}\left\|\boldsymbol{W}^\top \boldsymbol{X}^\top \boldsymbol{h}_u - \boldsymbol{y}^{(u)}\right\|_{\mathcal{F}}^2$. For a specific class $i \in \{1, 2, \cdots, c\}$, we have the gradient of $\ell^{(u)}$ w.r.t. $\boldsymbol{w}_i$ is

$$\frac{\partial \ell^{(u)}}{\partial \boldsymbol{w}_i} = \boldsymbol{X}^\top \boldsymbol{h}_u \boldsymbol{h}_u^\top \boldsymbol{X} \boldsymbol{w}_i - \boldsymbol{X}^\top \boldsymbol{h}_u \times y_i^{(u)}. \tag{D.2}$$

It is not hard to see from eq.(D.2) that the Hessian of $\ell^{(u)}$ with respect to $\boldsymbol{w}_i$ is

$$\nabla_{\boldsymbol{w}_i}^2 \ell^{(u)}(\boldsymbol{W}) = \boldsymbol{X}^\top \boldsymbol{h}_u \boldsymbol{h}_u^\top \boldsymbol{X}. \tag{D.3}$$

For classification tasks, the target $\boldsymbol{y}^{(u)}$-s are one-hot vectors representing the class of the node. Notice when we calculate cross entropy loss, we use softmax to normalize it before feeding it into the loss function. In the following for a vector $\boldsymbol{v} \in \mathbb{R}^c$ whose $i$-th entry is $v_i$, we define

$$\mathsf{softmax}(\boldsymbol{v})_i = \frac{\exp(v_i)}{\sum_{j=1}^c \exp(v_j)}. \tag{D.4}$$

Suppose the $u$-th node belongs to class $k$, then the cross entropy loss of the $u$-th node is defined as

$$\mathsf{CE}^{(u)}(\boldsymbol{W}) = -\log \mathsf{softmax}\left(\boldsymbol{h}_u^\top \boldsymbol{X} \boldsymbol{W}\right)_k \tag{D.5}$$

$$= -\boldsymbol{h}_u^\top \boldsymbol{X} \boldsymbol{w}_k + \log \sum_{j=1}^c \exp\left(\boldsymbol{h}_u^\top \boldsymbol{X} \boldsymbol{w}_j\right), \tag{D.6}$$

The gradient of $\mathsf{CE}^{(u)}$ w.r.t. $\boldsymbol{w}_i$ is

$$\frac{\partial \mathsf{CE}^{(u)}}{\partial \boldsymbol{w}_i} = -\boldsymbol{\delta}_{i,k} \times \boldsymbol{X}^\top \boldsymbol{h}_u + \frac{\boldsymbol{X}^\top \boldsymbol{h}_u \exp\left(\boldsymbol{h}_u^\top \boldsymbol{X} \boldsymbol{w}_i\right)}{\sum_{j=1}^c \exp\left(\boldsymbol{h}_u^\top \boldsymbol{X} \boldsymbol{w}_j\right)} \tag{D.7}$$

$$= \boldsymbol{X}^\top \boldsymbol{h}_u \mathsf{softmax}\left(\boldsymbol{h}_u^\top \boldsymbol{X} \boldsymbol{W}\right)_i - \boldsymbol{X}^\top \boldsymbol{h}_u \times y_i^{(u)}. \tag{D.8}$$

By taking another partial differentiation, we have

$$\frac{\partial^2 \mathsf{CE}^{(u)}}{\partial w_{p,i} \partial w_{q,i}} = \frac{\left(\boldsymbol{x}_p^\top \boldsymbol{h}_u\right)\left(\boldsymbol{x}_q^\top \boldsymbol{h}_u\right)\exp\left(\boldsymbol{h}_u^\top \boldsymbol{X} \boldsymbol{w}_i\right)}{\sum_{j=1}^c \exp\left(\boldsymbol{h}_u^\top \boldsymbol{X} \boldsymbol{w}_j\right)} - \frac{\left(\boldsymbol{x}_p^\top \boldsymbol{h}_u\right)\left(\boldsymbol{x}_q^\top \boldsymbol{h}_u\right)\left[\exp\left(\boldsymbol{h}_u^\top \boldsymbol{X} \boldsymbol{w}_i\right)\right]^2}{\left[\sum_{j=1}^c \exp\left(\boldsymbol{h}_u^\top \boldsymbol{X} \boldsymbol{w}_j\right)\right]^2} \tag{D.9}$$

$$= \left(\boldsymbol{x}_p^\top \boldsymbol{h}_u\right)\left(\boldsymbol{x}_q^\top \boldsymbol{h}_u\right)\left[s_i - s_i^2\right], \tag{D.10}$$

where $s_i = \mathsf{softmax}\left(\boldsymbol{h}_u^\top \boldsymbol{X} \boldsymbol{W}\right)_i$. Rewriting eq.(D.10) into matrix form, we have

$$\nabla_{\boldsymbol{w}_i}^2 \mathsf{CE}^{(u)}(\boldsymbol{W}) = (s_i - s_i^2)\boldsymbol{X}^\top \boldsymbol{h}_u \boldsymbol{h}_u^\top \boldsymbol{X}. \tag{D.11}$$

Comparing eq.(D.2) and eq.(D.8), we can notice that the gradient of MSE loss and CE loss are essentially the same except the term $\boldsymbol{h}_u^\top \boldsymbol{X} \boldsymbol{w}_i$ in eq.(D.2) is normalized by softmax in eq.(D.8). Additionally, by comparing the Hessian of MSE loss eq.(D.3) and the Hessian of CE loss eq.(D.11), we have the latter is a rescaled version of the former. These similarities intuitively explains why our method can be also used on CE loss.

## D.2 THE UNAVOIDABLE $\lambda^2$ IN THE RUNNING TIME

Lemma C.5 essentially states the following idea: even when two PD matrices are very similar (in terms of spectral approximation), if they are ill-conditioned, their square can be very different. In this subsection, we prove that the bound obtained in Lemma C.5 is strict up to constant factors by explicitly constructing a worst case. Notice that, in this subsection we discuss the issue of squaring spectral approximators in a broader context, thus in this subsection we possibly overload some symbols used before to simplify the notation.

First we consider another definition of approximation rate: for two PD matrices $\boldsymbol{\Sigma}$ and $\widetilde{\boldsymbol{\Sigma}}$, we define the approximation rate as

$$\psi\left(\boldsymbol{\Sigma}, \widetilde{\boldsymbol{\Sigma}}\right) = \max\left\{\left\|\boldsymbol{\Sigma}^{-1/2}\widetilde{\boldsymbol{\Sigma}}\boldsymbol{\Sigma}^{-1/2}\right\|, \left\|\widetilde{\boldsymbol{\Sigma}}^{-1/2}\boldsymbol{\Sigma}\widetilde{\boldsymbol{\Sigma}}^{-1/2}\right\|\right\}. \tag{D.12}$$

This definition is easier to calculate in the scenario considered in this subsection, and can be easily translated to the definition we used in Section 3: when $\epsilon = \psi\left(\boldsymbol{\Sigma}, \widetilde{\boldsymbol{\Sigma}}\right) \in (0,1)$, we have $\widetilde{\boldsymbol{\Sigma}} \approx_\epsilon \boldsymbol{\Sigma}$, and when $\psi\left(\boldsymbol{\Sigma}, \widetilde{\boldsymbol{\Sigma}}\right)$ is larger than 1 we don't have an approximation in the form defined in Section 3.

Let $\gamma > 1$ and $\delta \in (0,1)$ be real numbers. Define

$$\boldsymbol{\Sigma} = \begin{bmatrix} 1 & 0 \\ 0 & 1 \end{bmatrix} \begin{bmatrix} \gamma & 0 \\ 0 & 1 \end{bmatrix} \begin{bmatrix} 1 & 0 \\ 0 & 1 \end{bmatrix}, \tag{D.13}$$

and

$$\widetilde{\boldsymbol{\Sigma}} = \begin{bmatrix} \sqrt{1-\delta^2} & -\delta \\ \delta & \sqrt{1-\delta^2} \end{bmatrix} \begin{bmatrix} \gamma & 0 \\ 0 & 1 \end{bmatrix} \begin{bmatrix} \sqrt{1-\delta^2} & \delta \\ -\delta & \sqrt{1-\delta^2} \end{bmatrix}. \tag{D.14}$$

It's not hard to compute the error rate $\psi\left(\boldsymbol{\Sigma}, \widetilde{\boldsymbol{\Sigma}}\right) = \left\| \boldsymbol{\Sigma}^{-1/2} \widetilde{\boldsymbol{\Sigma}} \boldsymbol{\Sigma}^{-1/2} \right\| \approx \Theta(\gamma\delta^2)$. While if we consider the squared matrices, we have $\psi\left(\boldsymbol{\Sigma}^2, \widetilde{\boldsymbol{\Sigma}}^2\right) = \left\| \boldsymbol{\Sigma}^{-1} \widetilde{\boldsymbol{\Sigma}}^2 \boldsymbol{\Sigma}^{-1} \right\| \approx \gamma^2\delta^2$. In this case, $\psi\left(\boldsymbol{\Sigma}^2, \widetilde{\boldsymbol{\Sigma}}^2\right)$ is larger than $\psi\left(\boldsymbol{\Sigma}, \widetilde{\boldsymbol{\Sigma}}\right)$ by a factor of $\gamma$, which is the condition number of $\boldsymbol{\Sigma}$. When $\gamma$ is very large and $\delta$ is very small, we can have a small $\psi\left(\boldsymbol{\Sigma}, \widetilde{\boldsymbol{\Sigma}}\right)$ while large $\psi\left(\boldsymbol{\Sigma}^2, \widetilde{\boldsymbol{\Sigma}}^2\right)$, which .

**A More General Construction.** For a PD matrix $\boldsymbol{\Sigma}$ and its spectral approximation $\widetilde{\boldsymbol{\Sigma}}$, we call $\frac{\psi\left(\boldsymbol{\Sigma}^2, \widetilde{\boldsymbol{\Sigma}}^2\right)}{\psi\left(\boldsymbol{\Sigma}, \widetilde{\boldsymbol{\Sigma}}\right)}$ the Squared Error Rate. As noted above, in the worst case the squared error rate can be as large as the condition number of $\boldsymbol{\Sigma}$. However, the construction above is limited to $2 \times 2$ matrices. Now we construct a more general worst case of the squared error rate and perform a loose analysis. Although not rigorously proved, the construction and the analysis suggest the origination of large squared error rates: it approaches the upper bound (condition number) when the eigenspace of the two matrices are very well aligned but not exactly the same.

Let $\boldsymbol{A}$ be an ill-conditioned matrix with all eigenvalues very large except one eigenvalue equals to 1, and the eigenvalues of $\boldsymbol{B}$ are all closed to the eigenvalues of $\boldsymbol{A}$. Specifically, Let the SVD of $\boldsymbol{A}$ be $\boldsymbol{A} = \boldsymbol{U}\mathrm{diag}(\boldsymbol{\lambda})\boldsymbol{U}^\top$, where $\boldsymbol{\lambda} = \begin{bmatrix} \lambda_1 & \lambda_2 & \cdots \lambda_n \end{bmatrix}$, and suppose $\lambda_n = 1$ and $\lambda_k \gg 1, \forall k \leq n-1$. For simplicity we just let $\boldsymbol{A}$ and $\boldsymbol{B}$ have the same eigenvalues. Suppose the SVD of $\boldsymbol{B}$ is $\boldsymbol{B} = \boldsymbol{V}\mathrm{diag}(\boldsymbol{\lambda})\boldsymbol{V}^\top$ and let $\boldsymbol{\Lambda} = \mathrm{diag}(\boldsymbol{\lambda})$. We denote the one sided error rate of $\boldsymbol{A}$ and $\boldsymbol{B}$ by $\epsilon$, i.e. $\epsilon = \left\| \boldsymbol{A}^{-1/2}\boldsymbol{B}\boldsymbol{A}^{-1/2} \right\|$. We have

$$\epsilon = \left\| \boldsymbol{\Lambda}^{-1/2}\boldsymbol{U}^\top\boldsymbol{V}\boldsymbol{\Lambda}\boldsymbol{V}^\top\boldsymbol{U}\boldsymbol{\Lambda}^{-1/2} \right\| \tag{D.15}$$

$$= \left\| \boldsymbol{\Lambda}^{-1/2}\boldsymbol{W}\boldsymbol{\Lambda}\boldsymbol{W}^\top\boldsymbol{\Lambda}^{-1/2} \right\|, \tag{D.16}$$

where $\boldsymbol{W} = \boldsymbol{U}^\top\boldsymbol{V}$.

Since $\lambda_n = 1$ and for all $k \leq n-1$, $\lambda_k \gg 1$, we have

$$\boldsymbol{\Lambda}^{-\frac{1}{2}} = \begin{bmatrix} \lambda_1^{-1/2} & & & \\ & \lambda_2^{-\frac{1}{2}} & & \\ & & \ddots & \\ & & & \lambda_n^{-\frac{1}{2}} \end{bmatrix} \approx \begin{bmatrix} 0 & & & \\ & 0 & & \\ & & \ddots & \\ & & & 1 \end{bmatrix}, \tag{D.17}$$

and therefore

$$\epsilon = \left\| \boldsymbol{\Lambda}^{-1/2}\boldsymbol{W}\boldsymbol{\Lambda}\boldsymbol{W}^\top\boldsymbol{\Lambda}^{-1/2} \right\| \approx (\boldsymbol{W}\boldsymbol{\Lambda}\boldsymbol{W}^\top)_{n,n} = \sum_{k=1}^{n} \boldsymbol{W}_{k,n}^2 \lambda_k, \tag{D.18}$$

where $\boldsymbol{A}_{i,j}$ represents the $(i,j)$-th entry of a matrix $\boldsymbol{A}$.

For simplicity, we assume all $\lambda_k (k \leq n-1)$ are approximate equal, i.e. $\lambda_1 \approx \lambda_2 \cdots \approx \lambda_{n-1} = \gamma$. Then we have

$$\epsilon = \sum_{k=1}^{n} \boldsymbol{W}_{k,n}^2 \lambda_k \tag{D.19}$$

$$\approx \gamma \left[ \sum_{k=1}^{n-1} \boldsymbol{W}_{k,n}^2 + \frac{1}{\gamma} \boldsymbol{W}_{n,n}^2 \right] \tag{D.20}$$

$$\approx \gamma \sum_{k=1}^{n-1} \boldsymbol{W}_{k,n}^2 \tag{D.21}$$

$$= \gamma \left( 1 - \boldsymbol{W}_{n,n}^2 \right). \tag{D.22}$$

Recall $\boldsymbol{W} = \boldsymbol{U}^\top \boldsymbol{V}$, we have $\boldsymbol{W}_{n,n} = \boldsymbol{u}_n^\top \boldsymbol{v}_n$, where $\boldsymbol{u}_n$ and $\boldsymbol{v}_n$ are the $n$-th column of $\boldsymbol{U}$ and $\boldsymbol{V}$ respectively. Thus we have

$$\epsilon \approx \gamma \left[ 1 - \left( \boldsymbol{u}_n^\top \boldsymbol{v}_n \right)^2 \right]. \tag{D.23}$$

Now we can see the spectral approximation error $\epsilon$ is determined by two terms: the condition number $\gamma$ and the matchness of the eigenvectors corresponds to small eigenvalues, which is evaluated by $\left[ 1 - \left( \boldsymbol{u}_n^\top \boldsymbol{v}_n \right)^2 \right]$. **When $\gamma$ is very large but $\left[ 1 - \left( \boldsymbol{u}_n^\top \boldsymbol{v}_n \right)^2 \right]$ is small, $B$ can still be a spectral approximation of $A$.** For example if $(1 - \boldsymbol{u}_n^\top \boldsymbol{v}_n) \approx \gamma^{-1}$, we can get a spectral approximation error $\epsilon \approx 1$. However, after we square the matrices, the eigenvalues will also get squared, but eigenvectors remains unchanged. That will enlarge the spectral approximation error by a factor of $\gamma$, i.e. $\left\| \boldsymbol{A}^{-1} \boldsymbol{B}^2 \boldsymbol{A}^{-1} \right\| \approx \gamma^2 \gamma^{-1} = \gamma$, which becomes very large. In this case the squared error rate is $\gamma$, the condition number of $\boldsymbol{A}$.

### D.3 APPLYING GRAPH SPARSIFICATION IN EACH ITERATION

As mentioned in the main paper, unlike most existing work, in our algorithm we don't sparsify the graph in each training iteration. The proof of Lemma 5.1 (see Appendix C.3) suggests the reason why performing graph sparsification in each iteration can lead to suboptimal running time. In this subsection we illustrate this claim in detail. The intuition is that, in order to ensure convergence of the training, we require a small error rate in the gradient estimation, which is of the order $\epsilon^{1/2}$ as we have showed in Appendix C.3. It is acceptable for the SDD solver because the running time of the SDD solver only logarithmly depends on the error rate. However, if we sparsify the graph, to obtain an $\epsilon^{1/2}$ error rate we will need to sample $O(n_\lambda/\epsilon)$ edges, which grows linearly with $1/\epsilon$, and can be large especially when $\epsilon$ is very small. Below is a more detailed analysis.

Consider in the for-loop of Algorithm 1 we replace the $\hat{\boldsymbol{L}}$ by a sparsified version $\boldsymbol{L}' = \mathsf{Sparsify}_\omega \left( \hat{\boldsymbol{L}} \right)$, where $\omega \in (0, 1)$. Let $\boldsymbol{H}' = \boldsymbol{I} + \lambda \boldsymbol{L}'$. From Lemma 5.2, we have $\boldsymbol{H}' \approx_\omega \boldsymbol{H}$. Consider $E_1$ defined in Appendix C.3, it now becomes

$$E_1' = \left\| \boldsymbol{X}^\top \left[ \mathcal{S} \left( \boldsymbol{H}'^{-1} \boldsymbol{z} \right) - \boldsymbol{H}^{-2} \boldsymbol{z} \right] \right\| \tag{D.24}$$

$$\leq \sigma_{\max} \left( \boldsymbol{X} \right) \left\| \mathcal{S} \left( \boldsymbol{H}'^{-1} \boldsymbol{z} \right) - \boldsymbol{H}^{-2} \boldsymbol{z} \right\|. \tag{D.25}$$

Here we can not proceed by using the fact that $\mathcal{S}$ is a linear solver, because now $\mathcal{S}$ is not a linear solver for $\boldsymbol{H}$ but for $\boldsymbol{H}'$, thus we will have to split the error term again:

$$E_1' \leq \sigma_{\max} \left( \boldsymbol{X} \right) \left\| \mathcal{S} \left( \boldsymbol{H}'^{-1} \boldsymbol{z} \right) - \boldsymbol{H}'^{-2} \boldsymbol{z} \right\| + \sigma_{\max}(\boldsymbol{X}) \left\| \boldsymbol{H}^{-2} \boldsymbol{z} + \boldsymbol{H}'^{-2} \boldsymbol{z} \right\|. \tag{D.26}$$

Of the two terms on the right-hand side of eq.(D.26), the first one can be bounded with a similar method used in Appendix C.3, and the second term is bounded by

$$\sigma_{\max}(\boldsymbol{X}) \left\| \boldsymbol{H}^{-2} \boldsymbol{z} - \boldsymbol{H}'^{-2} \boldsymbol{z} \right\| \leq \sigma_{\max} \left( \boldsymbol{X} \right) \left\| \boldsymbol{I} - \boldsymbol{H}^{-1} \boldsymbol{H}'^{-2} \boldsymbol{H}^{-1} \right\| \left\| \boldsymbol{H}^{-2} \boldsymbol{z} \right\| \tag{D.27}$$

$$\leq \sigma_{\max} \left( \boldsymbol{X} \right) \omega \left\| \boldsymbol{H}^{-2} \boldsymbol{z} \right\| \tag{D.28}$$

$$\leq \sqrt{\frac{8}{\epsilon}} \kappa(\boldsymbol{X}) \lambda \omega \left\| \nabla \ell_i \left( \boldsymbol{w} \right) \right\|. \tag{D.29}$$

Therefore, in order to obtain $E_1' \leq \|\nabla \ell_i(\boldsymbol{w})\|$, we at least require $\omega \leq \frac{\epsilon^{-1/2}}{\sqrt{8}\kappa(\boldsymbol{X})\lambda} = O\left(\epsilon^{-1/2}\right)$. From Lemma 5.2, the number of edges in the sparsified graph $\omega$ error rate is $O(n_\lambda/\epsilon)$. A similar analysis can be also applied to $E_2$, and by repeating the proof in Appendix C.3, we obtain an overall running time bound

$$O(m + n_\lambda\lambda^2 d + d^3 + n_\lambda\epsilon^{-1}\log 1/\epsilon), \tag{D.30}$$

which, although eliminates the $m\left(\log\frac{1}{\epsilon}\right)^2$ term, introduces an extra $n_\lambda\epsilon^{-1}\log\frac{1}{\epsilon}$ term, and is usually worse than the original bound we derived in Theorem 5.1, especially when requiring a relatively small $\epsilon$.

That being said, although not very likely, when $\epsilon$ and / or $\lambda$ are large, it is possible that directly sparsifying the graph in each iteration is beneficial. Considering this, we can adopt a mixed strategy: when $n_\lambda\epsilon^{-1} \leq m\log(1/\epsilon)$, we sparsify the graph in each iteration, elsewise we don't. This leads to the following overall running time bound:

$$O\left[m + n_\lambda\lambda^2 d + d^3 + \min\left\{m\log\frac{1}{\epsilon}, n_\lambda\epsilon^{-1}\right\}\log\frac{1}{\epsilon}\right], \tag{D.31}$$

which is slightly better than the one we presented in Theorem 5.1.

### D.4 Possible Directions for Extending HERTA to More Complex Models

The assumption that $f(\boldsymbol{X}; \boldsymbol{W}) = \boldsymbol{XW}$ provides us with a convenience that the Hessian of this model is a constant matrix. Therefore in Algorithm 1 we only need to calculate the preconditioner $\boldsymbol{P}$ for one time. However, if $f(\boldsymbol{X}; \boldsymbol{W})$ is implemented by a non-linear network, then the Hessian will change by the time and might be hard to calculate, which will be a key challenge to using a more complex $f$. We note that this can be possibly addressed by constructing a linear approximation of $f$ using its Jaccobian. In each iteration, we can use the Jaccobian to replace the $\boldsymbol{X}$ used in current version of HERTA, and recalculate $\boldsymbol{P}$ at each iteration. Since the convergence is fast, we only need to recalculate the Jaccobian for a small number of iterations, so should not bring massive change to the running time of the algorithm.

The attention mechanism of TWIRLS in Yang et al. (2021b) is achieved by adding a concave penalty function to each summand of the the the $\text{Tr}\left(\boldsymbol{Z}^\top \hat{\boldsymbol{L}}\boldsymbol{Z}\right)$ term in eq.(4.1). For specific penalty functions, it might still possible to find a inner problem solver, as long as the problem stay convex. We note that this depends on concrete implementation of the penalty term used. Investigating how to fast solve the inner problem under various penalty functions should also be an important problem for future study.

### E Implementation Details

In the experiments, the datasets are loaded and processed using the DGL package Wang et al. (2019). We use the original inner problem solver in Yang et al. (2021b) since it is not computation bottleneck. It can be in principle replaced by any implementation of SDD solvers.

For the calculating the gradient of the preconditioned model, we presented a calculation method in Algorithm 1 which maintains the lowest computational complexity. In preliminary experiments, we tested this calculation method with using the autograd module in pytorch[3] and verified that they have the same output, and similar computational efficiency on real world datasets (again in practice this step is not a bottleneck). Therefore, we simply use pytorch autograd module to compute gradients in experiments.

Using the pytorch autograd module also enables us to apply HERTA on various loss functions and optimizers: we only need to perform the preconditioning and indicate the loss function. The gradient and optimization algorithm with be automatically realized by pytorch.

In the training loss experiments, in order to ensure a fair comparison and prevent confounding factors, we don't using any common training regularization techniques like weight decay or dropout. For each setting, we repeat the experiment with learning rates in $\{0.001, 0.01, 0.1, 1, 10\}$ and choose the

---

[3]https://pytorch.org/

trial which the training loss does not explode and with the lowest final training loss to report in the main paper.

In the test accuracy experiments, since for some tasks a regularizer is needed to obtain a normal test performance, we optionally allow an input dropout. Specifically, we do a grid search with input dropout rate in $\{0, 0.8\}$, learning rate in $\{0.001, 0.01, 0.1, 1, 10\}$, and $\lambda$ in $\{0, 1, 2, 10, 20, 100\}$. In comparison, we filtered out the trials with a too large and frequent oscillation, i.e. there are more than 10 epochs with validation set accuracy change more than $5\%$.

## F   ADDITIONAL EXPERIMENT RESULTS

In this section, we present additional experiment results.

### F.1   EXPERIMENTS WITH LARGER $\lambda$

All the experiments presented in the main paper are with $\lambda = 1$. In this subsection, we present results with $\lambda = 20$. See Figures 4 and 5 for results with MSE loss and CE loss respectively. The results supports our observation in the main paper that HERTA works consistently well on all settings.

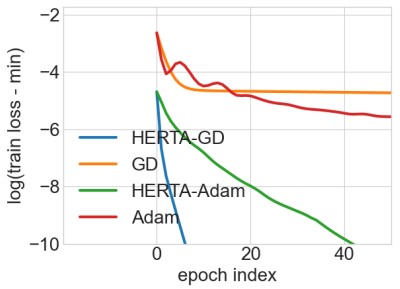 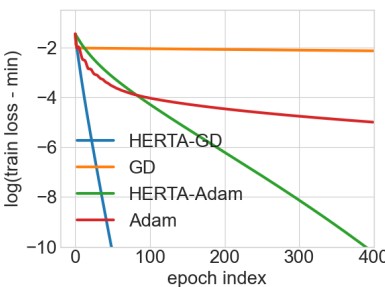

Figure 4: The training loss comparison between HERTA and standard optimizers on MSE loss with $\lambda = 20$. Dataset used from left to right: ogbn-arxiv, Pubmed.

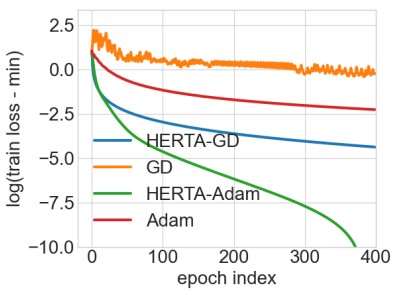 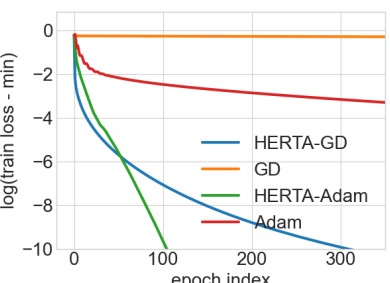

Figure 5: The training loss comparison between HERTA and standard optimizers on CE loss with $\lambda = 20$. Dataset used from left to right: ogbn-arxiv, Pubmed.

### F.2   ADDITIONAL TRAINING LOSS RESULTS ON CORA, CITESEER AND FLICKR

In this subsection we present training loss results on Cora and Citeseer, which are other citation datasets used in Yang et al. (2021b), as well as Flickr which is introduced by Zeng et al. (2019). See Figure 6 and Figure 9 for results with $\lambda = 1$ and $\lambda = 20$ on Cora respectively. See Figure 7 and Figure 10 for results with $\lambda = 1$ and $\lambda = 20$ on Citeseer respectively. See Figure 8 and Figure 11 for results with $\lambda = 1$ and $\lambda = 20$ on Flickr respectively. It is clear that the training loss on these datasets is consistent with our observation on other datasets.

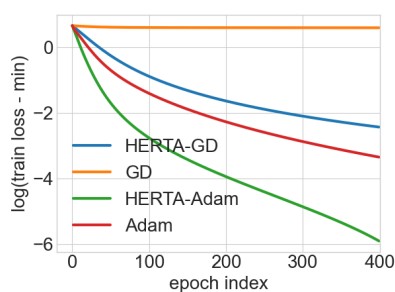
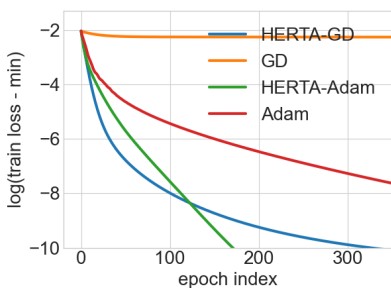

Figure 6: The training loss comparison between HERTA and standard optimizers on Cora with $\lambda = 1$. Left: CE loss. Right: MSE loss.

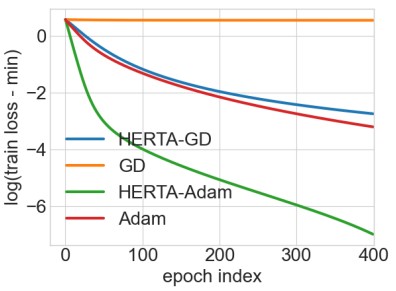
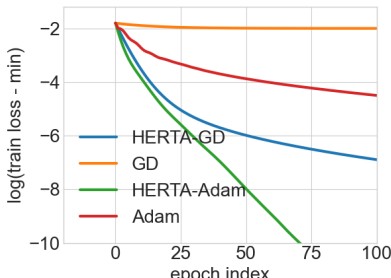

Figure 7: The training loss comparison between HERTA and standard optimizers on Cora with $\lambda = 1$. Left: CE loss. Right: MSE loss.

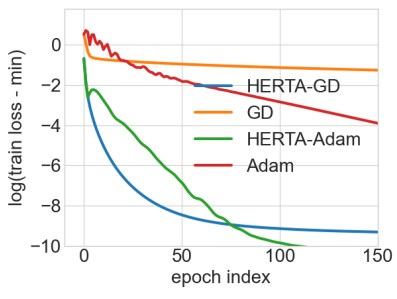
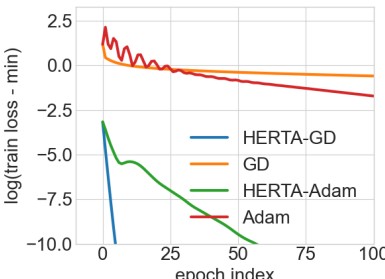

Figure 8: The training loss comparison between HERTA and standard optimizers on Cora with $\lambda = 1$. Left: CE loss. Right: MSE loss.

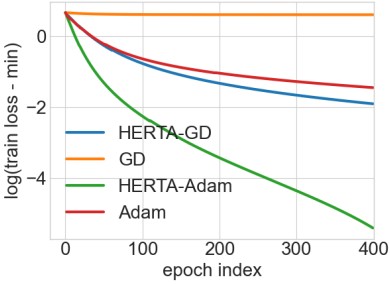
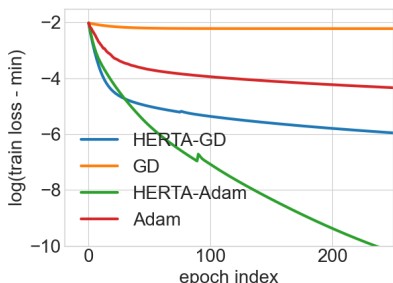

Figure 9: The training loss comparison between HERTA and standard optimizers on Cora with $\lambda = 20$. Left: CE loss. Right: MSE loss.

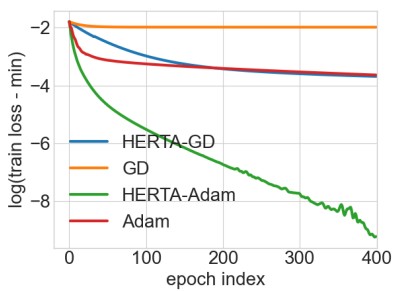 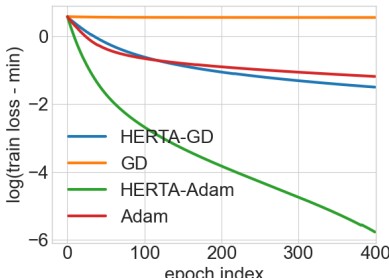

Figure 10: The training loss comparison between HERTA and standard optimizers on Citeseer with $\lambda = 20$. Left: CE loss. Right: MSE loss.

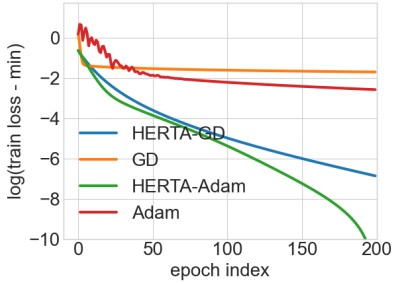 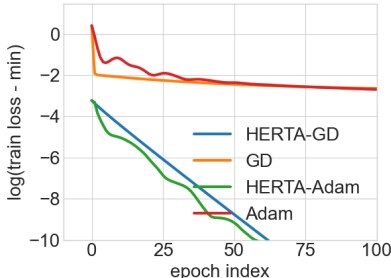

Figure 11: The training loss comparison between HERTA and standard optimizers on Cora with $\lambda = 1$. Left: CE loss. Right: MSE loss.

### F.3 ADDITIONAL TEST ACCURACY RESULTS

In this subsection we present test accuracy results on CE loss, as well as on other datasets, under the same setting as the main paper. See Figures 12 and 13 for the results with Cora and Citeseer respectively. See Figure 14 for the test accuracy comparison with CE loss on ogbn-arxiv, Pubmed and Flickr.

It is worth noting that Cora and Citeseer are both relatively small datasets (with number of nodes less than $4000$), and it seems that in terms of test accuracy, HERTA generally perform better with larger datasets. We argue that this comes from that fact that on small datasets, the optimization is not a major obstacle, and the standard optimizers can optimize them very fast as well. In this case, generalization more relies on regularization and implicit biases, which is not the focus of HERTA.

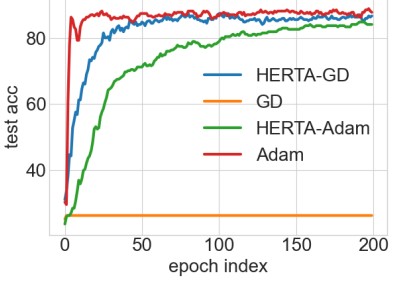 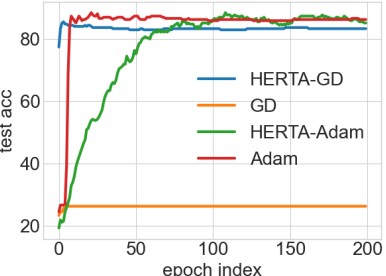

Figure 12: The test accuracy comparison between HERTA and standard optimizers trained on Cora. Left: CE loss. Right: MSE loss.

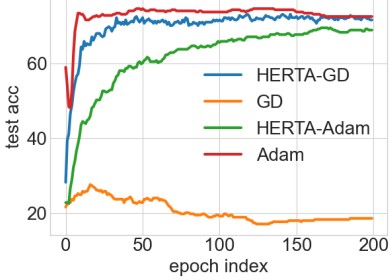 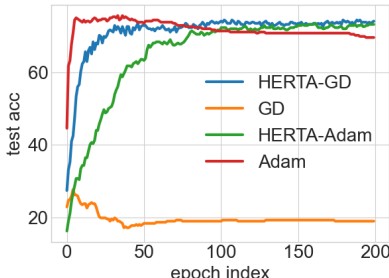

Figure 13: The test accuracy comparison between HERTA and standard optimizers trained on Citeseer. Left: CE loss. Right: MSE loss.

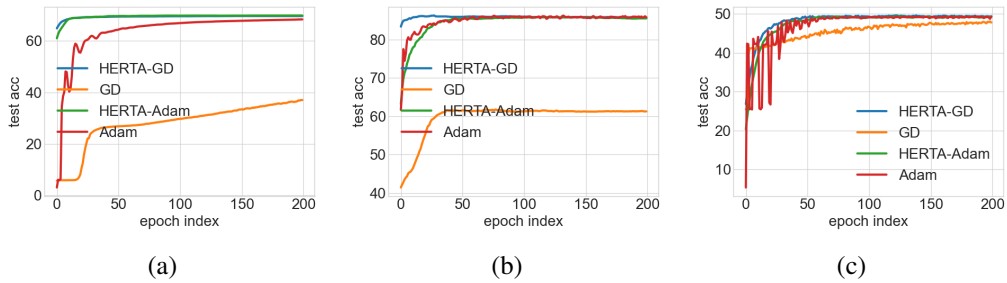

Figure 14: The test accuracy comparison between HERTA and standard optimizers trained with CE loss. (a) ogbn-arxiv; (b) Pubmed; (c) Flickr.

