# OpenReview forum: "HERTA: A High-Efficiency and Rigorous Training Algorithm for Unfolded Graph Neural Networks"
_ICLR.cc/2025/Conference — ICLR 2025 Conference Withdrawn Submission_

### Official Review · Reviewer_ccnj · 2024-10-31

**Soundness:** 3
**Presentation:** 3
**Contribution:** 2
**Rating:** 5
**Confidence:** 3

**Summary:**

The paper presents HERTA, an algorithm that optimizes the training of Unfolded Graph Neural Networks (GNNs), which are favored for their interpretability and flexibility but face scalability issues due to high training costs. HERTA addresses two key challenges: slow iteration rates and slow convergence. Unlike previous approaches that compromise interpretability for efficiency, HERTA achieves near-linear worst-case convergence while preserving the model's structure. The algorithm introduces a novel spectral sparsification technique that approximates the inverse of normalized and regularized Laplacian matrices, accelerating the training process without altering the underlying model. Experimental results on real-world datasets validate HERTA's ability to improve training speed and adapt to various loss functions and optimizers, showcasing its versatility and effectiveness for practical applications.

**Strengths:**

- The paper addresses a critical limitation in Unfolded Graph Neural Networks, which is high training costs, by proposing a novel training algorithm, HERTA, that achieves near-linear worst-case convergence without compromising the interpretability of the model.

- The work is theoretically robust and supported by empirical validation on real-world datasets in context of solving the training problem of Unfolded Graph Neural Networks

-  The paper is overall well-organized, with a clear problem statement and motivation for the research

**Weaknesses:**

- Minor issue with citation format; please ensure correct use of \cite and \citep commands throughout the paper.

-  The definition of the H-norm in Equation 5.2 is unclear. Providing a brief clarification would improve readability.

- Minor wording issue: the third line of Lemma 5.1 appears to be missing a 'for' in its description, which affects clarity.

- The paper’s focus is on Unfolded Graph Neural Networks, yet a comparison with other types of GNNs could further highlight the advantages of Unfolded GNNs over alternative architectures.

- Although HERTA is shown to outperform standard optimizers, it would be beneficial to include additional baseline methods from within the family of Unfolded Graph Neural Networks (papers cited in your work). This would provide a clearer comparison and better highlight HERTA’s specific advantages within this framework.

**Questions:**

- When comparing with standard GNNs, could you list the specific benefits of Unfolded GNNs, such as training speed, accuracy, interpretability?

- In the demonstration of Theorem 5.1, the phrase 'with a proper step size $\eta$ and number of iterations $T$ seems informal. Could you clarify or formalize this expression?

---

### Official Review · Reviewer_9nzQ · 2024-11-01

**Soundness:** 3
**Presentation:** 1
**Contribution:** 2
**Rating:** 5
**Confidence:** 3

**Summary:**

Current training algorithms for unrolled GNNs suffer from slow iterations and slow convergence rates, and solutions for these issues often require the modification of the original model.

The authors bridge this gap by introducing HERTA, an algorithm to efficiently train Unfolded GNNs, without modifying the original unrolled model.

Rigorous convergence and complexity analysis are provided, along with various convincing experiments on real-world datasets.

**Strengths:**

The HERTA algorithm is built on thorough and rigorous theory.

The technical results are detailed and extensive.

The authors conduct experiments on various real-world datasets, and the presented results display convincing performance.

**Weaknesses:**

I will provide here some suggestions on the presentation as well as technical questions.

* I find Theorem 1.1 to be misplaced in the Introduction. It references its formal version and an objective which come much later. We do not have prior exposure to the manipulated objects or the task at hand at that point. More generally speaking, my suggestion would be to limit the level of technicalities in the introduction which hinders the exposure and induces too many references to later sections (lines 66 to 83). Instead of stating results of complexity analysis, I think it would rather be more useful to informally discuss the core ideas behind the HERTA algorithm (which is presented as the core contribution in the introduction) and/or present the Unfolded GNN that is considered in this work (Eq. 4.1).

* It is also somewhat frustrating to go through convergence results and complexity analysis before even being exposed to the algorithm or at least being presented its core ideas. The HERTA algorithm is presented as being the core contribution in the introduction but we have to wait until Section 5.4 (page 8) to finally be exposed to it for the first time. Concretely speaking, I would provide an overview of the algorithm, explaining the general ideas, before discussing Theorem 5.1.

* Overall, in my opinion, the organisation of the paper is somewhat unusual. This is supported, for example, by the numerous references to later sections or paragraphs (for instance in lines 66, 67, 82, 247, 257) which makes the exposure more difficult. As mentioned earlier, I believe the paper would gain in clarity if the HERTA algorithm is introduced (at least informally) earlier on, and before delving into the theoretical analysis.

* I find the formulation "it is not hard to see that [...]" to be overused (lines 161, 216, 803, 1247). Perhaps you should provide brief explanations instead ?

* In line 166, it is not clear whether "optimizing the following objective" refers to its minimization or maximization. In my opinion, it should be specified that the objective is being minimized, as explicited later on.

* The authors do not provide the code for the implementation of their algorithm or their experiments, and it is not clear in the paper if it would be made public upon acceptance. I think the authors should include a statement about code availability.

* HERTA is essentially an optimization procedure for solving the bi-level optimization problem TWIRLS [Yang et al.,2021]. From a graph signal processing point of view, we are then trying to learn a signal representation $Z^*$ on a given graph $\hat{L}$, parametrized by weights $W$, such that it minimizes a distance $l$ to the label signals given by $Y$. The term $Tr(Z^\top \hat{L} Z)$ in the energy function enforces a smooth representation of the signals on the graph. In the context considered in this paper, the graph represented by its Laplacian $\hat{L}$ is considered given. Would it be possible to extend TWIRLS and HERTA to the graph learning paradigm, where in addition to learning a proper signal representation $Z^*$, we are also seeking to learn the graph topology, in other words to learn both $Z^*$ and $\hat{L}$ ? How would this reflect on HERTA ? Perhaps could you discuss this potential extension in the future work section ?

* In most unrolled optimization procedures, the original algorithm's hyperparameters (such as the step-size or the regularization term $\lambda$) are learned in a data-driven fashion. From my understanding, this was not pursued in this work, could you comment on this choice, and how this would reflect on HERTA ?

**Questions:**

Please refer to the Weaknesses section.

**Details Of Ethics Concerns:**

Not applicable.

---

### Official Review · Reviewer_FH29 · 2024-11-03

**Soundness:** 2
**Presentation:** 2
**Contribution:** 2
**Rating:** 3
**Confidence:** 4

**Summary:**

The paper introduces HERTA, a new Graph Neural Network that is designed based on Unfolded GNN.

The approach is designed to accelerate the training process and to ensure convergence.

Moreover, a new spectral sparsification method is also proposed.

**Strengths:**

HERTA achieves good results on some benchmark datasets like ogbn-arxiv compared with standard optimizers.

**Weaknesses:**

- In Section 1, the author claims that the running time is $\widetilde{O}((m+nd)(\log{\frac{1}{\epsilon}})^2+d^3)=\widetilde{O}((m+nd)(\log{\frac{1}{\epsilon}})^2)$ as $ d << m$ in practice, where the feature dimensionality is $d$ and the edge number is $m$.

    However, this does not mean that $\widetilde{O}(d^3)<\widetilde{O}(m)$.
    Moreover, in most benchmark datasets, $d^3 >> m$. Take PubMed for example (the paper also conducts experiments on PubMed), $d = 500$ and $m = 44324$, we have $d^3 > 2000m$. Even in a large dataset, ogbn-arxiv, with a relatively small $d = 128$, and a large $m = 1,166,243$, it also holds that $d^3 \approx 1.8m$.

    This shows that the estimation of the running time of the approach is inappropriate.

- Moreover, no experiment on time and space burden is conducted to validate the efficiency of the approach.

- In Theorem 1.1, the number of large eigenvalues of the graph Laplacian is assumed as $O(n/\lambda^2)$.
    While in Definition 5.1, $n_\lambda$ is roughly the number of large eigenvalues, i.e., $O(n/\lambda^2)$.
    It's not true that $n_\lambda\rightarrow n$ as $\lambda\rightarrow\infty$

- In Definition 5.1, $n_\lambda$ roughly represents the number of eigenvalues of $\hat{L}$ which are of order $\Omega(\lambda^{-1})$.
    While the author also claims that $n_\lambda$ represents the number of eigen-directions for which the Laplacian regularizer $\frac{\lambda}{2}\text{Tr}(Z^\top\hat{L}Z)$ is large, where $Z\in\mathbb{R}^{n\times c}$ is the optimization variable.
    What is the relationship between the eigenvalues of $\hat{L}$ and that of $Z^\top\hat{L}Z$?
    I believe that they are not the same.

- The train loss of the approach is displayed in the form of $\log(\text{train loss}-\text{min})$, which enlarges the change of the loss (as the result is negative for most of the time).
    Moreover, the paper only shows the relative change of each method, the exact minimum value is not shown.
    This adds to the concern of the convergence of the method.

- In Appendix C, (C.3) is wrongly derived:
    Firstly, $g^{(t)}$ and $\nabla l(\omega^{(t)})$ in which should be exchanged. (otherwise (C.5) and (C.8) will not hold)
    Secondly, the derivation from the condition to (C.3) only holds for $l_1$-norm.

**Questions:**

- In Section 3, $||M||$ for a matrix $M$ is mentioned as its operator norm.
    What does that mean?
- Tiny Questions (as the main purpose is to accelerate) (while the paper claims that HERTA has excellent generalization properties)
  - There is no accuracy or variance to show the effectiveness of the approach.
  - The performance of the approach is not compared with SOTA baselines.

---

### Official Review · Reviewer_Zgpp · 2024-11-04

**Soundness:** 3
**Presentation:** 4
**Contribution:** 3
**Rating:** 5
**Confidence:** 4

**Summary:**

This paper introduces HERTA, a novel training algorithm for Unfolded Graph Neural Networks (GNNs) that aims to solve scalability issues while preserving interpretability. The algorithm achieves nearly linear time training guarantees and introduces a new spectral sparsification method for normalized and regularized graph laplacians. The authors demonstrate HERTA's effectiveness through theoretical analysis and empirical validation on real-world datasets.

**Strengths:**

1. Provides a novel spectral sparsification method for normalized/regularized Laplacians
2. Shows promising empirical results on larger datasets
3. Works with different loss functions and optimizers
4. Provides a solid foundation for future work in this direction

**Weaknesses:**

1. The main idea (on lines 211-217) behind the effective Laplacian dimension ($n_\lambda$) is that it represents how many eigenvalue directions "matter significantly" in the energy function. Smaller $n_\lambda$ means fewer “important directions” to consider, which in turn gives less computational complexity. This is explained via the closed form solution in Eqn 4.6 .
When the eigenvalue $\lambda_i$ is much smaller than $1/\lambda$, then the effect of the Laplacian is dominated by the $I$ (identity matrix). Larger $n_\lambda$ means more work during training as there is a need to preserve graph structure in more directions.

But the problem arises where for their theoretical guarantees, their algorithm requires $O(n/\lambda^2$ “large eigenvalues”, which implies a high effective dimension. This creates a contradiction because the authors require a low effective dimension for efficiency, but their algorithm's guarantees need a high effective dimension.
Please elaborate on how the algorithm's performance changes with different distributions of eigenvalues and provide examples of real-world graphs where this condition is satisfied.

2. Also, if we were to consider $\lambda$ to be an “absolute constant” (as written in lines 80-81), then the Big-Oh notation would reduce to $O(n)$ and not be $O(n/\lambda^2)$. In the experiments, the authors set $\lambda$ to 1 (and 20 in their appendix), which again means $O(n)$ in essence. So, you would need a linear fraction of your eigenvalues to be large eigenvalues. Is this practical?
In real-world scenarios, you typically find very few large eigenvalues and the majority of eigenvalues as small (closer to 0).
Please discuss the typical distribution of eigenvalues in real-world graphs and how this affects the performance of their algorithm.

3. In theorem 5.1, equation 5.1, the authors have a $d^3$ term in the final time complexity analysis equation. In their informal Thm 1.1 (lines 72-73), they say “In practice, the node feature dimensionality d is generally much smaller than the graph size m, in which case, the running time of HERTA is …” And they ignore the $d^3$ term.

If we take a closer look at this assumption, there arise several problems with it. Modern ML models often use very high-dimensional features. E.g. Word2vec embeddings have dimensions $d$ around 100-300, BERT embeddings have $d=768$, and LLMs with $d=1024$ or larger.
Let’s say for d=300, $d^3$ is 27M, for d=768, $d^3$ is approx. 453 M and for d=1024, it's around 1B!

Even if $d \le m$, $d^3$ can dominate. For m = 10k, d=100, $d^3$ is 1M! The cubic growth makes d^3 significant in the analysis and cannot simply be ignored.
Please justify and provide conditions for when $d^3$ can be ignored. The only time is when $m >> d$, but this would be way too restrictive a condition for modern applications.

In my opinion, this term should remain and be analyzed properly to say when each term dominates and discuss the trade-offs with feature dimensionality.

- Provide a more detailed analysis of when each term in the complexity bound dominates.
- Discuss the trade-offs between feature dimensionality and algorithm performance.
- Consider revising your informal theorem statement to include the $d^3$ term and provide a more nuanced discussion of when it can be ignored.
- Provide empirical results showing how the algorithm's performance scales with feature dimensionality.

**Experimental Section related issues.**

1. The paper doesn't properly justify why these specific datasets (Cora, Citeseer, Pubmed, ogbn-arxiv, and Flickr) were chosen beyond mentioning that some were used in Yang et al. (2021b). There's no discussion of dataset characteristics (like sparsity, feature dimensions, etc.) that would help understand how these choices stress-test different aspects of the algorithm.

2. There needs to be some analysis that shows the distribution of “large eigenvalues” in the graphs that have been chosen. This would help get more insights and better understand the theoretical bounds proposed on large eigenvalues as well. It can be the case that the theoretical bounds aren’t tight enough to explain why your method gets faster convergence.

3. The experiments only compare against gradient descent and Adam optimizers. Other relevant baselines like SGD with momentum, RMSprop, or specialized GNN optimizers are missing. This limits understanding of the proposed method’s relative performance against the full spectrum of optimization approaches.

4. In Section 6.1, Figures 1 and 2, can you please explain why a log scale was chosen and how it affects the interpretation of the results. There is no justification given in the paper as to why this was necessary.

5. Despite claiming theoretical time complexity benefits, there are no direct runtime comparisons or scaling experiments showing how HERTA's performance changes with graph size, feature dimensionality, or other relevant parameters. It would be nice, even on a synthetic graph dataset, to see how increasing the graph size (and also feature dimensionality) will affect this method’s convergence runtimes.

6. As mentioned in Section F3 “on small datasets (Cora and Citeseer with <4000 nodes), HERTA's test accuracy advantages are less clear since optimization isn't a major obstacle." This suggests that this method may not have a universal utility.

7. There are no ablation experiments showing the relative importance of different components of HERTA (like the preconditioner construction, spectral sparsification, etc.). This makes it hard to understand which aspects are crucial for performance improvements. Could it be that the graphs considered already have a good gap between their first and second eigenvalues and thus converging well? Please provide more empirical evidence.

8. The theoretical analysis focuses primarily on a simple linear implementation of f(X;W) = XW, leaving more complex model architectures unexplored. In the empirical studies, it would be interesting to at the least show performance of the method on more complex deep learning architectures for f.

9. There is a limited analysis of hyperparameter effects and sensitivity analysis. A systematic stability analysis would be very useful. Also, how does one go about selecting the parameters?

These weaknesses suggest that while HERTA shows promising results, more comprehensive experimental evaluation would strengthen the paper's empirical claims and our understanding of when and why the method works best.

- Can you provide a justification for the choice of datasets and discuss their relevant characteristics (sparsity, feature dimensions, etc.) that make them suitable for evaluating HERTA?

- Please include an analysis of the eigenvalue distribution for the chosen graphs and discuss how this relates to the theoretical bounds.

- Consider expanding the comparison to include other relevant optimizers like SGD with momentum, RMSprop, and specialized GNN optimizers.

- Explain the rationale behind using a log scale in Figures 1 and 2, and discuss how this affects the interpretation of the results.

- Include direct runtime comparisons and scaling experiments

**Questions:**

Despite these issues, the paper has some valuable contributions. I recommend that the authors provide more comprehensive experiments and discuss the limitations and complexity properly. The theoretical contradictions must be resolved and clear practical guidelines on how to use this method should be presented.
The core ideas are promising, but the current execution has too many significant weaknesses to warrant acceptance, even as a weak accept. The combination of theoretical inconsistencies and experimental limitations suggests the work needs substantial revision before it's ready for publication.

---

### Note · Authors · 2024-11-25

**Comment:**

We have decided to withdraw our paper after careful consideration of the reviewers' feedback. We sincerely thank the reviewers for their valuable insights and constructive suggestions, which we deeply appreciate.

**Withdrawal Confirmation:**

I have read and agree with the venue's withdrawal policy on behalf of myself and my co-authors.